# Alcam-a and Pdgfr-α are essential for the development of sclerotome-derived stromal cells that support hematopoiesis

Emi Murayama [1,2,3] ✉, Catherine Vivier[1,3], Anne Schmidt [1,3] & Philippe Herbomel [1,3]

Mesenchymal stromal cells are essential components of hematopoietic stem and progenitor cell (HSPC) niches, regulating HSPC proliferation and fates. Their developmental origins are largely unknown. In zebrafish, we previously found that the stromal cells of the caudal hematopoietic tissue (CHT), a niche functionally homologous to the mammalian fetal liver, arise from the ventral part of caudal somites. We have now found that this ventral domain is the sclerotome, and that two markers of mammalian mesenchymal stem/stromal cells, Alcam and Pdgfr-α, are distinctively expressed there and instrumental for the emergence and migration of stromal cell progenitors, which in turn conditions the proper assembly of the vascular component of the CHT niche. Furthermore, we find that trunk somites are similarly dependent on Alcam and Pdgfr-α to produce mesenchymal cells that foster HSPC emergence from the aorta. Thus the sclerotome contributes essential stromal cells for each of the key steps of developmental hematopoiesis.

Hematopoietic stem and progenitor cells (HSPCs) are multipotent precursors that have the self-renewal capacity and continuously replenish all mature blood cells throughout the life span. In zebrafish as in mammalian development, they initially emerge from the dorsal aorta of the embryo through an endothelial hematopoietic transition (EHT)[1–4]. Then they enter the bloodstream to reach their first niche, the caudal hematopoietic tissue (CHT)[5], where they expand and undergo their first multi-lineage differentiation[6]. Thus the CHT in fish is the hematopoietic homolog of the fetal liver in mammals. In our previous work, we showed that the CHT niche was mainly composed of a transient venous plexus arising from the primitive caudal vein, and mesenchymal stromal cells interconnecting as well as lining the branches of this venous plexus[5,7]—a structure basically similar to that of mammalian hematopoietic niches[8].

We then discovered that the progenitors of these stromal cells initially arose from the ventral part of the caudal somites through an epithelial–mesenchymal transition (EMT)[7]. These stromal cell progenitors (SCPs) then emigrated further ventrally, in close interaction with the endothelial cells sprouting from the primitive caudal vein that

would make up the venous plexus, the other main component of the niche[7]. In the present study, we have further characterized the process of SCP emergence and emigration from the caudal somites and identified important molecular actors in this process.

The sclerotome compartment of somites, which has been more studied in the trunk, also lies at their ventromedial side and undergoes EMT and subsequent emigration, to produce axial cartilage and bone around the notochord[9,10], as well as other derivatives such as mural cells around blood vessels[11–14] and tendon cells that will link muscles to the axial skeleton[15]. Here we have also addressed the relationship of the sclerotome with the ventral cell clusters of the caudal somites that give rise to the SCPs.

Activated leukocyte cell adhesion molecule (Alcam or CD166) is a large cell surface glycoprotein of the immunoglobulin superfamily that mediates adhesion through homophilic interactions in various tissues[16], and heterophilic interactions with CD6 at the interface of T cells and antigen-presenting cells[17], and with galectin-8 in the extracellular matrix[18]. While expressed in a wide variety of tissues, Alcam is often found in cell subsets involved in dynamic growth and/or

[1]Institut Pasteur, Department of Developmental & Stem Cell Biology, Paris 75015, France. [2]INSERM, Paris 75013, France. [3]CNRS, UMR3738, Paris 75015, France.
✉ e-mail: emur@pasteur.fr

migration processes including neural development[19], pharyngeal pouch formation[20], angiogenesis[18], kidney development[21], and tumor progression[22]; it is also found in mesenchymal stromal cells of bone marrow and fetal liver[23–25]. The short cytoplasmic tail of Alcam was shown to interact indirectly with the actin cytoskeleton via actin-binding proteins such as ezrin and syntenin-1[26]. These interactions of the cytoplasmic domain dynamically regulate the clustering of Alcam molecules at the cell surface and the strength of the homophilic and heterophilic interactions[27,28].

Platelet-derived growth factor receptor α (PDGFR-α) is a highly conserved receptor tyrosine kinase. The binding of PDGFs to PDGF receptors induces their dimerization, which unlocks their tyrosine kinase activity and results in the autophosphorylation of specific tyrosine residues, which then act as binding sites for intracellular Src homology 2 (SH2) domain-containing signaling molecules[29,30]. PDGFR-α mediated signaling regulates various processes of embryonic development and organogenesis, notably the proliferation then migration, and differentiation of specialized mesenchymal cells in various organ anlages[31]. PDGFR-α is also a marker of bone marrow stromal cells, that has been widely used to purify them[32,33].

Here we identify Alcama and Pdgfr-α as two essential transmembrane proteins in the process of SCP emergence from somite epithelial cells, their subsequent migration to become stromal cells, and the resulting structure and functionality of the CHT niche. We further show that the somite ventral clusters giving rise to them coincide with the caudal sclerotome and that Alcama and Pdgfr-α are similarly important for the emergence from the trunk somites of sub-aortic mesenchymal cells that foster the emergence of HSPCs from the aorta floor.

## Results

### Alcama is required for stromal cell development from somites

We previously described that cell clusters containing SCPs appeared at the ventral side of caudal somites by 21–22 hpf[7]. More precisely, the relative timing of this process among caudal somites derives from the timing of somite formation, which occurs in the rostrocaudal direction, with one new somite forming every 30 min. To monitor their subsequent maturation, it is convenient to number somites dynamically, calling S1 the most recently formed somite, S2 its rostral neighbor formed 30 min earlier, and so on[34]. Within this framework, we found that the ventral clusters (VC) first become apparent by live video-enhanced DIC/Nomarski (VE-DIC) microscopy in the transition from somite maturation stage S4–S5, hence 2.5 h after somite formation (Fig. 1A). During cluster formation, the cells located on the ventral side of the somites adopt a rosette-like structure with a small cavity in the center (Fig. 1A-a and Supplementary Movie 1). To visualize the subsequent emigration of SCPs from these VCs, we previously used TCF-driven fluorescent reporters expressed in all somite cells, which were merely inherited by the emigrating SCPs[7]. In the present study, to analyze the fate of these SCPs, we first used the Tg(pax3a:eGFP) and Tg(ET37:eGFP) lines[35]. Tg(pax3a:eGFP) labeled somite cells weakly, then the VCs more strongly, as well as the migrating SCPs (Supplementary Movie 2) up to 2.5 dpf. The Tg(ET37:eGFP) line also highlights somite cells weakly[35], then starts to label SCPs more strongly as they start to emigrate from the somites (Supplementary Fig. 1A). We also aimed to create a new Tg line that would label SCPs more specifically and throughout their development. We first searched for genes that seemed specifically expressed in the VCs in the ZFIN whole-mount in situ hybridization (WISH) databank. Our subsequent study by WISH of candidate genes between 23 and 48 hpf led us to target the chondroitin sulfate proteoglycan 4 (cspg4) gene, which we found to be expressed in the VCs from somite maturation stage S6 onwards (Supplementary Fig. 1B). We thus generated a BAC-based Tg line expressing the GAL4 transcription factor (TF) from the cspg4 regulatory regions. This TgBAC(cspg4:Gal4) line crossed with a

Tg(UAS:RFP) reporter line faithfully recapitulates the cspg4 gene expression pattern spatiotemporally (Fig. 1B, Supplementary Fig.1C–E), although expression is somewhat mosaic—a typical limitation of Gal4/UAS expression systems in zebrafish. With the exception of the notochord, in the caudal region RFP expression highlights the VCs and their derivatives. Upon live imaging of Tg(cspg4:Gal4; UAS:RFP; ET37:eGFP) embryos, we noticed that by 29 hpf, RFP⁺GFP⁺ cells were migrating towards not only the ventral but also the dorsal side (Supplementary Fig. 1C). The latter cell population migrated along the medial side of the somites towards the notochord, suggesting chondro-progenitor and/or tenocyte fates, hence sclerotomal nature[9,10,15]. Ventral-wards migrating SCPs gave rise to stromal cells throughout the CHT by 38 hpf, and some migrated even further ventrally to become fin mesenchymal cells[35] (FMCs) in the caudal fin (Fig. 1B-b). These three fates of somite VC-derived cells are recapitulated in Supplementary Fig. 1F.

To understand how somite epithelial cells compartmentalize to give rise to the VCs and their derivatives, we also searched for cell adhesion molecules expressed specifically in the VCs. We found that the alcama gene (coding for activated leukocyte cell adhesion molecule a, aka CD166), was expressed in the VCs more strongly than in the bulk of the somites at 23 hpf, and its expression was already observed, albeit slightly, on the ventral side of somites from the S2/S3 stage, i.e. before cluster formation (Fig. 1C). Immunofluorescence allowed us to first detect the Alcama protein at the very core of the clusters as they first appeared morphologically, i.e. at somite stage S5 (Fig. 1D-a, b), suggesting post-transcriptional control of alcama expression at earlier somite maturation stages. Upon somite maturation to S6–S8, Alcama protein signals then gradually propagated to delineate cell boundaries within the clusters (Fig. 1D-a, b). We then detected the Alcama protein at joints between VC-derived cells migrating both ventral- and dorsal-wards (Fig. 1D-c). We examined the impact of alcama knock-down by using an antisense morpholino oligonucleotide (MO) that completely blocked translation of the alcama transcript in vivo (Supplementary Fig. 1G, H). The resulting morphant embryos had a globally normal development, somite VCs formed and cells began to emigrate from them. However, injection in Tg(pax3a:eGFP;TCF:nls-mCherry) embryos revealed that while in control embryos, SCPs initially migrated as strings of 2–3 cells interconnected by a rather large contact surface (Fig. 1E-a, c), which immunostaining showed to be enriched in Alcama protein (Fig. 1D-c), in alcama morphants, adhesion between migrating SCPs was reduced to a very small surface at the tip of an elongated cell protrusion (Fig. 1E-b, d). We next crossed our TgBAC(cspg4:Gal4;UAS:RFP) line with Tg(UAS:Lifeact-eGFP)[36], in which Lifeact-eGFP allows to highlight F-actin in vivo, so as to visualize actin dynamics and cellular projections during SCP emergence and migration. Lifeact-eGFP⁺ VC-derived cells were clearly less numerous in alcama morphants by 26 hpf (Fig. 1F)—a finding confirmed in the Tg(ET37:eGFP) background (Fig. 1G). This reduced number was later mirrored by a similarly (2.7-fold) lower number of SCPs after migration into the CHT by 30 hpf (Fig. 1G), and a clearly lower number of both stromal cells in the CHT and FMCs by 46 hpf (Fig. 1H).

### Alcama regulates the migration behavior of SCPs via F-actin

Time-lapse imaging of migrating Lifeact-GFP⁺ SCPs in alcama morphants revealed a characteristic cell morphology compared to SCPs in control embryos (Fig. 2A). Filopodial projections in migrating leader SCPs usually occur mainly at the leading edge, whereas they were 50% more numerous (Supplementary Fig. 2A-a) and more randomly scattered around the migrating cell in alcama morphants (Fig. 2B). In addition, even though they were slightly longer on average in alcama morphants (Supplementary Fig. 2A-b), unlike in control embryos their length did not correlate with their position relative to the direction of migration (Fig. 2C). A further quantitative analysis of confocal live images showed that (i) a high Lifeact-GFP (F-actin) signal was present

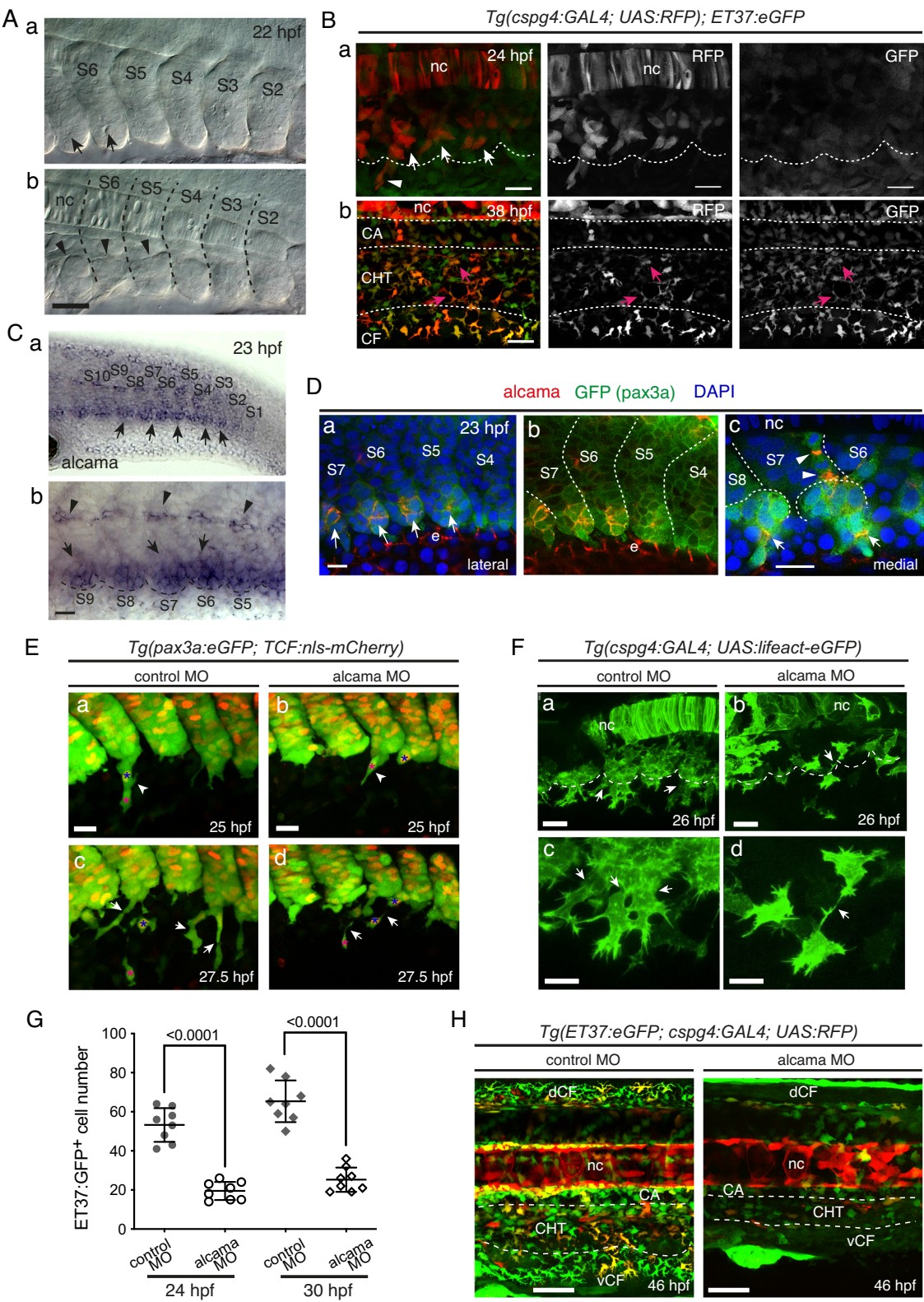

throughout migrating SCPs in alcama morphants whereas the F-actin signal was observed predominantly at the leading edge in controls (Supplementary Fig. 2B), and (ii) the surface area of migrating SCPs in alcama morphants was much larger than in controls (Supplementary Fig. 2C). These changes in morphology and F-actin distribution correlated with a strong defect in SCP migration, as their linear distance of migration in alcama morphants was less than half that in the control group (Fig. 2D). Later on, by 46 hpf, this migration defect had

translated in a much thinner distribution of ET37:eGFP+ stromal cells in the CHT (Fig. 1H).

To gain more insight into the implication of Alcama in SCP migration, we took advantage of the *Tg(cspg4:Gal4)* driver line to express specifically in these cells three different forms of Alcama fused to eGFP at its C-terminus—a WT full-length Alcama (Alcama-FL-eGFP), and two mutant forms: (i) Alcama-ΔN-eGFP, lacking both N-terminal Ig-like V-type domains that are known to mediate homophilic cell

**Fig. 1 | SCP development and requirement for Alcama. A** VE-DIC microscopy of developing caudal somites at 22 hpf, rostral to the left, dorsal to the top. Somites are numbered S2–S6 according to their maturation stage. Lateral (**a**) and more medial (**b**) focal planes show the central lumen (**a**, arrows) formed within the ventral somite clusters, and the dorsal border of the clusters (**b**, arrowheads). Dashed lines indicate intersomitic boundaries. Scale bar, 40 μm. **B** Confocal projections of live *Tg(cspg4:Gal4; UAS:RFP; ET37:eGFP)* embryos at 24 (**a**) and 38 (**b**) hpf. **a** Arrows and arrowhead point at VC cells and an emerging SCP, respectively; the dotted line outlines the ventral border of caudal somites. Magenta arrows in (**b**) point at RFP/GFP double-positive cells in the CHT. Scale bars, 20 μm. **C** WISH for alcama at 23 hpf. Wide-area image of the tail (**a**) and magnified view of the ventral part of somites S5–S9 (**b**). Arrows and arrowheads point at somite VCs and cells prefiguring the horizontal myoseptum, respectively; a dotted line outlines the somite ventral border; scale bar, 20 μm. **D** Immunostaining for Alcama (red), pax3a:eGFP (green), together with DAPI (blue) at 23 hpf. **a** Overlay of red, blue and green signals; arrows point at Alcama labeling in the somite VCs. **b** Same image as **a** without DAPI staining; e epidermis. **c** At a deeper focal plane, strong Alcama signals were detected at contacts between cells migrating dorsal-wards (arrowheads) and ventral-wards (arrows); dashed lines delineate the VCs and somite borders. Scale bars, 20 μm. **E** Confocal maximum projections of control or alcama MO-injected *Tg(pax3a:eGFP; TCF:nls-mCherry)* embryos at 25 (**a**, **b**) and 27.5 (**c**, **d**) hpf. (**a**, **b**) Arrowheads show SCPs beginning to emigrate; magenta and blue asterisks point to the same cells as in (**c**) and (**d**), respectively. **c**, **d** Arrows point at intercellular junctions of migrating SCPs. *n* = 12 for each condition from four independent experiments. Scale bars, 25 μm. **F** Confocal projections of control or alcama MO-injected *Tg(cspg4:Gal4; UAS:Lifeact-eGFP)* at 26 hpf. Dashed lines indicate the somite ventral borders. Arrows point at connections between migrating SCPs. Magnified images are shown in (**c**) and (**d**). *n* = 9 for each condition from three independent experiments. Scale bars, 20 μm. **G** Quantification of ET37:eGFP+ cells in the ventral tail of live control and alcama MO-injected embryos, at 24 and 30 hpf over a 5-somites width (*n* = 8 embryos each; mean ± SD; two-tailed Student's *t*-test, Source data are provided as a Source Data file). At 24 hpf, all GFP+ cells ventral to the notochord were counted; at 30 hpf, all GFP+ cells ventral to the somites were counted. **H** Confocal projections of control or alcama MO-injected *Tg(ET37:eGFP; cspg4:Gal4; UAS:RFP)* embryos at 46 hpf. Dashed lines delineate the CHT. *n* = 10 for each condition from a single experiment. Scale bars, 50 μm. nc notochord, CA caudal artery, CHT caudal hematopoietic tissue, vCF and dCF ventral and dorsal caudal fin.

adhesion[37] and (ii) Alcama-ΔPDZ-eGFP, in which we introduced in the short cytoplasmic domain of Alcama three amino acid changes predicted to suppress the binding of PDZ domain-containing proteins (Supplementary Fig. 2D), such as syntenin[26]. These different forms were inserted downstream of a UAS promoter, and the constructs were injected in *Tg(cspg4:Gal4;UAS:RFP)* embryos, leading to specific but mosaic expression of the GFP-linked mutant forms of Alcama among VC-derived cells. Expression of Alcama-FL-eGFP in the VCs did not perturb SCP emergence and migration (Supplementary Fig. 2E, Supplementary Movie 3). In contrast, SCPs expressing Alcama-ΔN-eGFP appeared disconnected from their neighbors within the VCs, and soon developed intense filopodial dynamics while still within the somite (Fig. 2Ea–c, Supplementary Movie 4), but most often with no resulting ventral-wards migration towards the CHT, whereas internal control neighboring RFP+GFP- cells migrated normally (Fig. 2E-d). Alcama-ΔPDZ-eGFP+ VC cells and their SCP derivatives displayed much fewer filopodia compared to WT or Alcama-ΔN-eGFP expressing cells; they still showed some GFP enrichment at cell-cell contacts, but no clear polarity (Fig. 2Fa–c, Supplementary Movie 5). Like Alcama-ΔN-eGFP expressing cells, Alcama-ΔPDZ-eGFP+ SCPs barely migrated to the ventral side, compared with (RFP+GFP−) SCPs in the same embryo (Fig. 2F-d). In contrast, the dorsal-wards migration of VC-derived GFP+ cells was rather retained in both cases, as well as for Alcama-FL-eGFP expressing cells (Fig. 2G and Supplementary Fig. 2F, G). In vivo tracking of GFP+ cells in both Alcama-ΔN-eGFP and -ΔPDZ-eGFP expressing embryos confirmed that the cells that egressed ventrally from the VCs then wandered around and migrated only shortly toward the ventral side, whereas the dorsal-wards migration was similar to that of control or Alcama-FL-eGFP expressing cells (Fig. 2H, I). Statistical analysis of their fates confirmed that the phenotype of GFP+ mutant cells was predominantly 'immobile' or 'dorsal migration', with occasional 'ventral migration' limited to a short distance (Fig. 2J). Altogether, these data show that Alcama is cell-autonomously involved in SCP emergence and migration.

## Alcama and PDGFR-α deficiency similarly affect SCP development

A prominent feature of alcama morphant and dominant-negative mutant phenotypes was the inhibition of SCP ventral migration. Therefore, we investigated the gene expression of receptors for cytokines and growth factors that might be involved in SCP migration. We found that *pdgfr-α* was specifically expressed in the somite VCs, starting by somite maturation stage S6, and then in the migrating cells derived from them (Fig. 3F-a, b). Therefore we hypothesized a possible cross-talk between Alcama and Pdgfr-α. It is known that PDGF signaling leads to an increase in AKT and ERK phosphorylation[38,39]. We first investigated the impact of *alcama* knockdown on pERK signaling in the caudal region at 26 hpf. The number of pERK+ cells was reduced to 36% of control level in the CHT of alcama morphants, while it recovered to almost the control level in the rescued group (Fig. 3A). Then, we investigated the impact of Pdgfr-α deficiency, using a previously validated MO against *pdgfra*[40]. First, we analyzed the colocalization of pERK-positive cells with SCPs marked by pax3a-GFP in alcama and pdgfra morphants. While in control embryos we could observe pERK+GFP+ migrating SCPs, in alcama and pdgfra morphants the overall number of pERK+ cells was reduced, and no pERK signal was observed among migrating SCPs (Supplementary Fig. 3A). Time-lapse imaging of SCP emergence in pdgfra morphants by 23–24 hpf revealed that the cohesion of the VCs was compromised, with groups of cells often delaminating together from the somite (Supplementary Fig. 3B), and then several SCPs per somite cluster or cell group starting to emigrate simultaneously (Fig. 3B), yet with a clearly reduced migration efficiency relative to WT embryos. Quantification showed that the numbers of CHT stromal cells and FMCs originating from SCPs were reduced 2.4-fold and 2.0-fold, respectively, in pdgfra morphants by 40 hpf (Fig. 3C), similar to what we previously found for alcama morphants (Fig. 1G, H). Live imaging also showed that like in alcama morphants, the extent of ventral-wards migration of SCPs by that time was much reduced in the pdgfra morphants (Supplementary Movie 6). This impacted the co-migration of endothelial cells that made the caudal venous (CV) plexus, resulting in a much thinner plexus, often reduced to a single convoluted tube, in apposition to the overlying caudal artery, which could locally show a reduced diameter (Fig. 3D).

We then attempted to identify PDGF ligands that would activate Pdgfr-α signaling in the VCs and migrating SCPs. We performed pERK immunostaining on embryos overexpressing either of three different PDGFs which we had found by WISH to be more specifically expressed in the ventro-caudal area (Supplementary Fig. 3C). All three ligands activated pERK signaling, hence likely Pdgfr-α signaling, but the effect of Pdgfaa and Pdgfab was statistically more significant than Pdgfbb (Supplementary Fig. 3D).

## PDGFR-α drives SCP migration through PI3K signaling

PI3K signaling is known to be activated downstream of various growth factor receptors including PDGF receptors[41] following their activation. Pharmacological inhibition of PI3K isoforms using AS605240 (inhibiting PI3K isoforms γ, α), LY294002 (inhibiting isoforms α, β, δ), or CAL101 (inhibiting isoform δ) revealed that the migration of SCPs was compromised in embryos treated with LY294002, and more so with AS605240, resulting in a 3.5-fold reduction in stromal cell numbers by

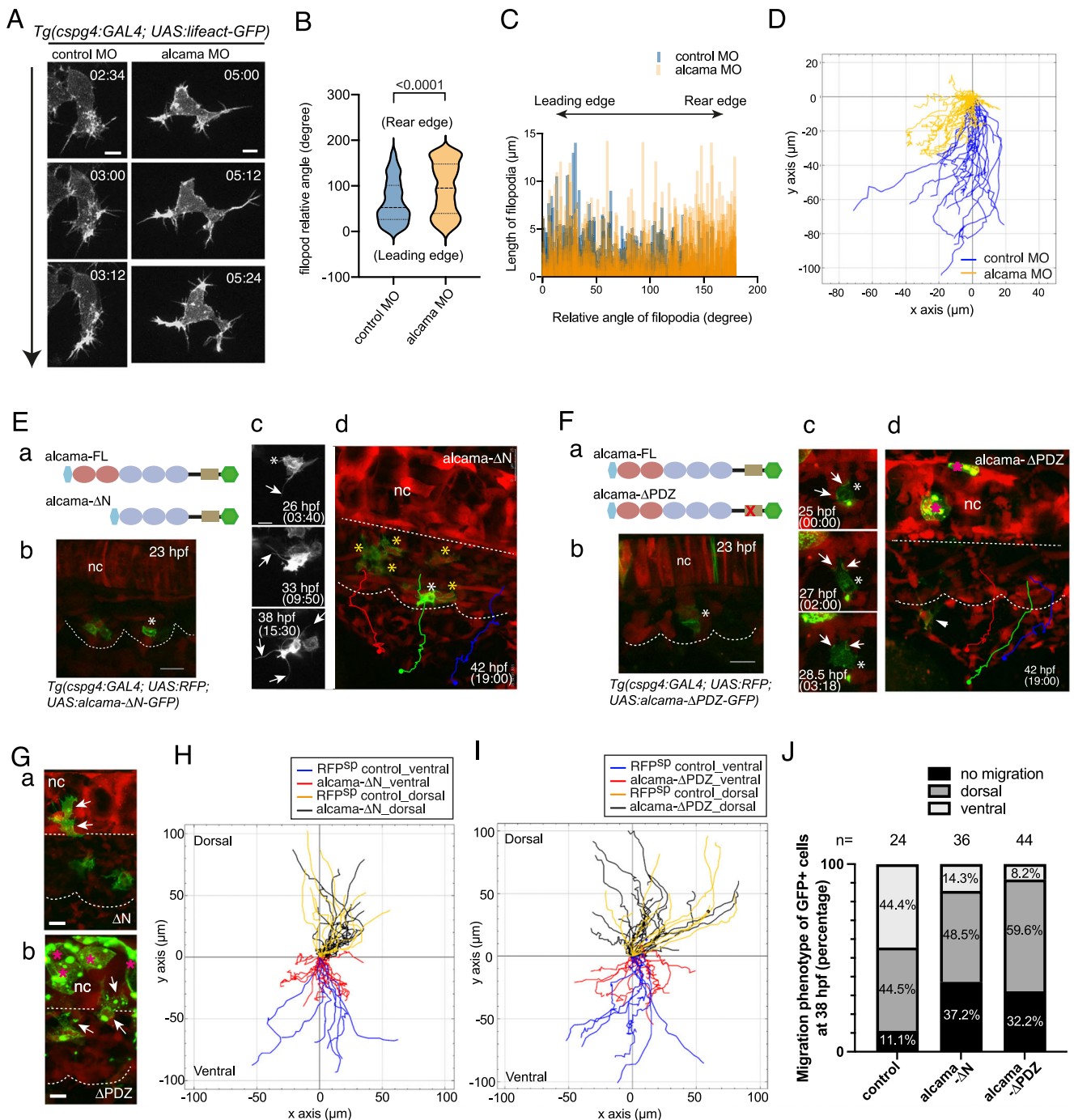

40 hpf (Fig. 3E), whereas pax3a-GFP^high neural crest cells migrated with only a slight delay (Supplementary Fig. 3E, blue arrows). To monitor the migration of SCPs and other VC-derived cells over longer intervals, AS-treated embryos were fixed and labeled with a pdgfra in situ probe at stages encompassing the entire migration process. In the control group, dorsal-wards migration of VC-derived cells had already started by 23 hpf, whereas no migration was observed in the AS-treated embryos (Fig. 3F-a, c). By 36 hpf, SCP migration had not started yet in the AS-treated group, in which SCPs accumulated at the zone of their emergence, while it was already complete in the control group (Fig. 3F-b, d).

Since the involvement of PI3K in SCP migration was revealed, we then mutated two tyrosine residues in the intracellular domain of Pdgfr-α that are responsible for the binding of class IA (α/β/δ) PI3K to Pdgfr-α following its activation and autophosphorylation[38] (Fig. 4A),

and we cloned the mutated or WT pdgfra ORF followed by a tandem HA-tag downstream of a hsp70 promoter[42] (hsp70:pdgfra-ΔPI3K-HA or hsp70:pdgfra^WT-HA, respectively). Transient transgenics resulting from injection of the hsp70:pdgfra-ΔPI3K-HA or hsp70:pdgfra^WT-HA construct in *Tg(pax3a:eGFP)* embryos were heat-shocked at 19–20 hpf, then imaged in vivo from 23 hpf, or fixed for immunofluorescence at the same time point. The anti-HA antibody signals sometimes showed a mosaic distribution, but were sufficiently present in the targeted tissue of hsp70:pdgfra-ΔPI3K-HA and hsp:70:pdgfra^WT-HA embryos at 24 hpf (Supplementary Fig. 4A). Heat-shocked hsp70:pdgfra-ΔPI3K-HA embryos typically showed a less clear outline of the somite VCs than controls. In addition, 38% of them showed multiple SCPs simultaneously initiating migration from one somite, reflecting the lack of leader cell, and 20.5% had at least one somite showing an 'overflow' phenotype in which the VC collectively delaminated into the ventro-

**Fig. 2 | Alcama regulates SCP migration. A** Time-lapse confocal images of a migrating SCP leader cell from control or alcama MO-injected *Tg(cspg4:Gal4; UAS:Lifeact-GFP)* embryos around 28 hpf. The time after initiation of SCP migration is shown in the upper right corner of each panel. The time lag between control and alcama morphant SCP is because the latter tend to take more time to fully egress from the somite. Arrow indicates the direction of migration. Scale bars, 10 μm. **B** Violin plot of filopodia angles relative to the direction of migration for leader SCPs in control and alcama morphant embryos (*n* = 7 SCPs for each group, from 3 independent experiments; median ± SD; two-tailed Mann–Whitney test). **C** Filopodia length as a function of their relative angles for leader SCPs in control and alcama morphants. **D** Overlay of individual tracks of control and alcama-deficient SCPs during their migration toward the CHT. (x,y) coordinates of the SCPs' starting points at somite borders are set to (0,0) and negative or positive values on the *y*-axis indicate ventral-wards or dorsal-wards migration, respectively. $n_{Exp}$ = 4, $n_{embryos}$ = 6, $n_{tracked\ cells}$ = 19 and 17 for control or alcama morphants, respectively. **E, F a** Schematic representation of Alcama-FL-eGFP, Alcama-ΔN-eGFP (**E-a**) and Alcama-ΔPDZ-eGFP (**F-a**) mutant constructs. Pink and blue ovals indicate the V-type and C2-type Ig-like domains, respectively; the beige rectangle represents the short cytoplasmic domain. In (**F-a**), the red x indicates the mutated PDZ binding motif. Light blue and green hexagons represent the signal peptide and GFP, respectively. **E-b** Appearance of Alcama-ΔN-eGFP⁺ SCPs at 23 hpf. White asterisk indicates the same cell as shown in (**E-c, d**). Dotted lines delineate somite and notochord borders. Scale bar, 20 μm. **c** Morphological evolution over 15.5 h of the Alcama-ΔN-eGFP⁺ SCP shown in (**E-b**) (see also Supplementary Movie 4). Scale bar, 10 μm. **E-d** End point of the time-lapse imaging shown in (**E-b, c**) and Supplementary

Movie 4. GFP⁺ SCPs marked by yellow asterisks did not migrate out of the somites, whereas RFP⁺/GFP⁻ WT SCPs showed a normal emigration pattern exemplified by the three colored trajectories. A straight dotted line indicates the notochord lower border. **F** (**b**) Appearance of Alcama-ΔPDZ-eGFP expressing VC cell (asterisk) at 23 hpf, followed over time in (**c**) (see also Supplementary Movie 5). Scale bar, 20 μm. **c** Arrows point at large contact areas between the GFP⁺ cell and RFP⁺ cells. **d** End point of the time-lapse imaging of the same embryo; the GFP⁺ SCP followed up in (**b, c**) had disappeared by 42 hpf, while another such cell displayed short-distance migration (arrowhead). RFP⁺/GFP⁻ (WT) SCPs showed a normal emigration pattern indicated by three colored trajectories. Straight and curvy dashed lines delineate the ventral border of notochord and somites. **G** Dorsal-wards migration of Alcama-ΔN-eGFP and -ΔPDZ-eGFP expressing cells (arrows) by 27 hpf. Scale bars, 10 μm. Magenta asterisks in (**F-d**) and (**G-b**) label GFP⁺ notochord cells. **H** Overlay of individual tracks of control (RFP⁺/GFP⁻ SCPs; orange and blue lines) and Alcama-ΔN-eGFP expressing SCPs (black and red lines). $n_{Exp}$ = 5, $n_{embryos}$ = 7, $n_{tracked\ cells}$ = 29 and 17 for Alcama-ΔN-eGFP⁺ and control RFP⁺/GFP⁻ SCPs, respectively. **I** Overlay of individual tracks of control (RFP⁺/GFP⁻ SCPs; orange and blue lines) and Alcama-ΔPDZ-eGFP⁺ SCPs (black and red lines). $n_{Exp}$ = 4, $n_{embryos}$ = 5, $n_{tracked\ cells}$ = 32 and 17 for Alcama-ΔPDZ-eGFP⁺ and control SCPs, respectively. **J** Frequency histogram of D/V migration patterns of Alcama-ΔN-eGFP⁺ and -ΔPDZ-eGFP⁺ cells compared with GFP⁺ cells of *Tg(cspg4:Gal4;UAS:GFP)* embryos at 38 hpf. The number of embryos analyzed was 24 (WT), 36 (Alcama-ΔN), and 44 (Alcama-ΔPDZ) from three independent experiments. nc notochord. Source data for **B, C, D H, I**, and **J** are provided as a Source Data file.

caudal cavity, reminiscent of the phenotype observed above in pdgfra morphants (Supplementary Fig. 4B); these cell populations tended to disperse over time but did not go through the normal migration process (Fig. 4B-b, c). In embryos overexpressing the pdgfra^WT-HA construct, more apoptotic bodies were detected than in WT embryos, but no abnormalities were observed in the VCs and emigration of SCPs (Fig. 4B-a). Interestingly, the morphology of migrating SCPs in pdgfra-ΔPI3K expressing embryos was quite similar to that observed in alcama morphants, with long and numerous filopodia and the cell's main axis often orthogonal to the ventral-wards direction of migration (Fig. 4C, Supplementary Movie 7), and with increased 3D cell surface area and volume (Supplementary Fig. 4C, D). Importantly, the number of VC-derived GFP⁺ cells was clearly more decreased for those migrating ventral-wards (SCPs) than for those migrating dorsal-wards (DMCs) (Fig. 4D), again as previously observed upon Alcama deficiency. The total number of SCPs at 36 hpf in the CHT of pdgfra-ΔPI3K expressing embryos was also strongly reduced (Fig. 4E), as in Alcama- or Pdgfra-deficient embryos.

We then checked whether the application of CRISPR/Cas9 technology to the alcama and pdgfra genes would confirm our findings. To mimic the genetic modification introduced by alcama and pdgfra MOs, gRNAs were designed to delete the promoter of the *alcama* gene (Supplementary Fig. 5A-a) and the intron2–exon3 splice junction of the *pdgfra* gene (Supplementary Fig. 5B-a). Genomic PCR and subsequent sequencing on individual alcama gRNA injected embryos indicated *alcama* promoter deletion in 8/12 embryos (Supplementary Fig. 5A-b). A similar analysis on pdgfra-gRNA injected embryos indicated *pdgfra* intron2–exon3 splice junction deletion in 10/16 embryos (Supplementary Fig. 5B-b left panel), and cDNA analysis on pooled pdgfra-gRNA injected embryos confirmed exon3 deletion in the resulting mRNA (Supplementary Fig. 5B-b right panel). The phenotypic consequences were analyzed in the *Tg(pax3a:eGFP; kdrl:ras-mCherry)* background. Relative to control embryos, alcama and pdgfra crispants both showed a greatly reduced number of pax3:GFP⁺ stromal cells in the CHT at 48 hpf (Supplementary Fig. 5C), and a malformed CV plexus often reduced to a single large convoluted tube (Supplementary Figs. 5D, E), similar to what we previously described for the morphants. Thus alcama and pdgfra crispants confirmed our previous findings using alcama and pdgfra morphants.

## Caudal somite ventral clusters coincide with the sclerotome

Since our live imaging revealed that the somite VCs from which SCPs originate also gave rise to cells migrating dorsal-wards towards the notochord, it suggested an overlap or identity of somite VCs with the sclerotome. We examined this point through a double WISH for the sclerotome marker *nkx3.1*[15] and RFP expressed in *Tg(cspg4:Gal4; UAS:RFP)* embryos. This revealed a complete coincidence of both stainings in the tail, hence the identity of the somite VCs with the sclerotome (Fig. 5A). We further examined the *Tg(ola-twist:Gal4; UAS:Kaede)* line, which has been used to trace sclerotome-derived cells in zebrafish. We found that Kaede expression in the tail at 24 hpf indeed covered a large subset of the nkx3.1/cspg4:Gal4 expression domain, starting only a bit later in somite maturation, by stage S8 (Fig. 5B). We then photoconverted Kaede in single expressing cells at the ventral border of the four anterior-most caudal somites at 23 hpf (which at that time represent somite maturation stages S8–S11), and tracked their fate up to 48 hpf (Fig. 5C, D). Three main fates were equally represented—stromal cells of the CHT (33%), DMCs (31%), and muscle fibers (31%), while the FMC fate was more marginal (5%) (Fig. 5E). This cell fate analysis thus confirmed our previous findings, but added the muscle fiber category, reminiscent of the pioneer study by Morin-Kensicki et al.[9] who found that the posterior part of each ventro-medial cell cluster gave rise to some muscle cells in addition to the typical sclerotome fates (Fig. 5F). Only SCP fated cells underwent mitosis during the observed period (4/14), and they did so on their way to or within the CHT; therefore all photoconverted cells showed only one fate during the tracked period, suggesting that cell fate within the VCs may already be restricted by somite stage S8 (the earliest stage at which Kaede was sufficiently expressed for photoconversion-based cell tracking).

## Alcama and Pdgfr-α expression are regulated by sclerotomal transcription factors

Our WISH analysis revealed that other TFs considered as sclerotome markers within the somites, such as Pax9, Snai2, Twist1a, and Twist1b were expressed in the VCs of caudal somites from somite stage S5/S6, and also quite earlier—from S1—for Snai2 (Fig. 6A). Therefore, we investigated if these TFs regulate the expression of *alcama* and *pdgfra* during SCP development. To this end, we first isolated promoter regions of the *alcama* (3.3 kb) and *pdgfra* (3.2 kb) genes from BACs and

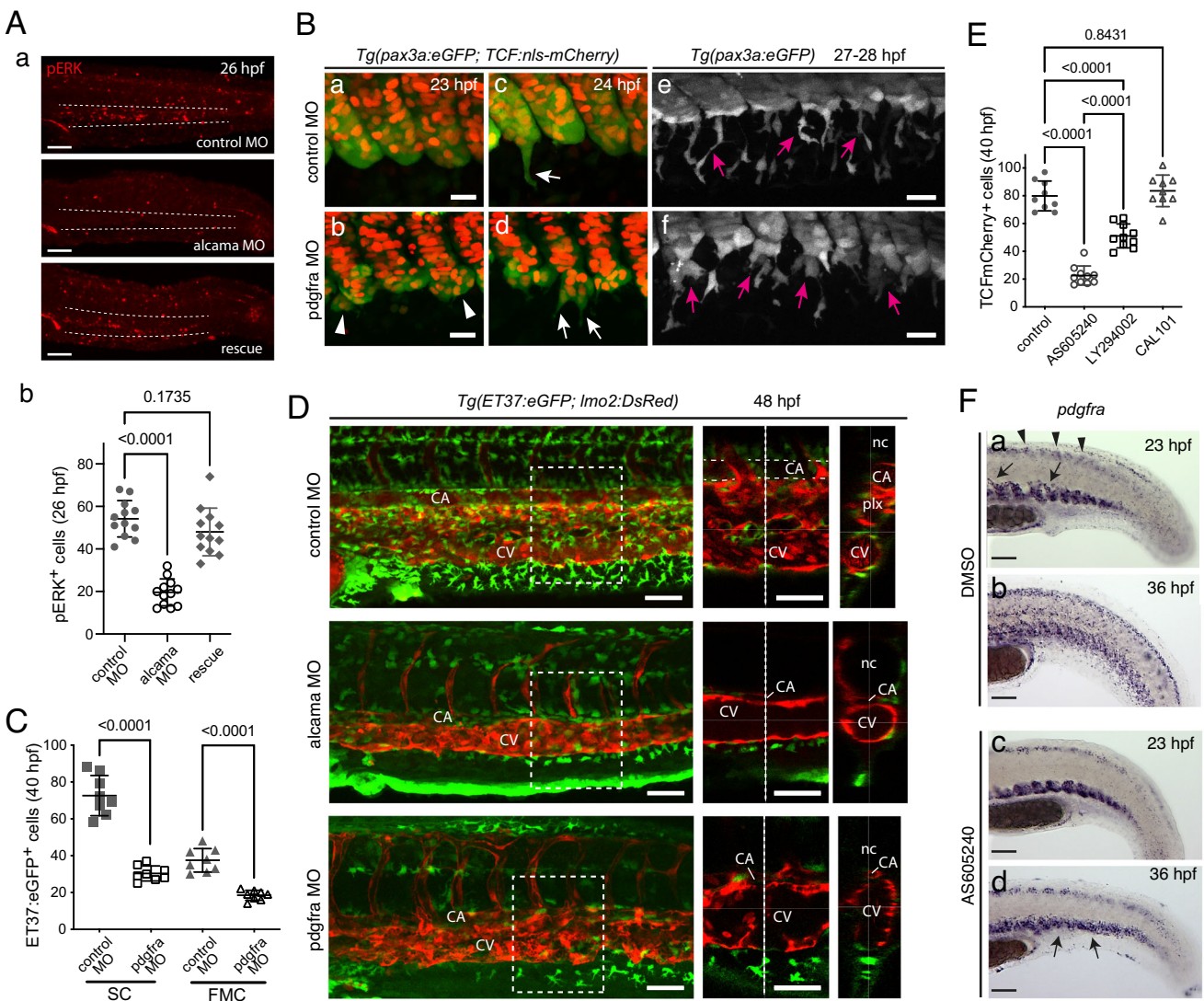

**Fig. 3 | Pdgfr-α/PI3K signaling regulates SCP migration. A** (**a**) Immunofluorescence for pERK at 26 hpf in the tail region of embryos injected with control MO, alcama MO, or alcama MO + alcama mRNA ("rescue"); *n* = 12 embryos from two independent experiments. Dashed lines indicate the CHT border. **b** Quantification of **a** (*n* = 12 embryos for each group; mean ± SD; one-way ANOVA followed by Dunnett's multiple comparison test). **B** Confocal projections of live control (**a**, **c**, **e**) or pdgfra (**b**, **d**, **f**) MO-injected *Tg(ET37:eGFP; TCF:nls-mCherry)* embryos from 23 to 28 hpf. Arrowheads point at cohesion defects observed in the VCs of pdgfra MO-injected embryos. White and magenta arrows indicate emerging (**c**, **d**) and migrating (**e**, **f**) SCPs, respectively; *n* = 9 for each condition from three independent experiments. Scale bars, 25 μm. **C** Quantification of ET37:eGFP⁺ stromal cells and FMCs in control and pdgfra MO-injected embryos at 40 hpf (*n* = 8 for each group, from the same experiment; mean ± SD; two-tailed Student's *t*-test). **D** Confocal maximum projections of control, alcama, and pdgfra MO-injected *Tg(ET37:eGFP;*

*lmo2:DsRed)* embryos at 48 hpf. An enlarged view of the area enclosed by the dashed square is shown in the middle panel (single slice image), where a dotted vertical line shows the position of the orthogonal projection (transverse section) shown in the right panel; *n* = 6 for each condition from three independent experiments. Scale bars, 50 μm. **E** Quantification of TCF:nls-mCherry⁺ stromal cells in the CHT of control embryos (*n* = 9), or embryos treated from 20 hpf with AS605240 (*n* = 10), LY294002 (*n* = 10), or CAL101 (*n* = 9) at 40 hpf (mean ± SD; one-way ANOVA followed by Tukey's multiple comparison test). **F** WISH for pdgfra at 23 and 36 hpf in 0.2% DMSO (control) and AS605240-treated embryos. Arrows and arrowheads in (**a**) indicate VC cells migrating toward the notochord and dorsal mesenchymal cells at 23 hpf, respectively. Arrows in **d** show SCPs slightly dispersed around the VCs at 36 hpf. *n* = 12 for each condition from two independent experiments. CA caudal artery, CV definitive caudal vein, Plx venous plexus, nc notochord. Source data for **A-b**, **C**, and **E** are provided as a Source Data file.

cloned them into the pGL3 Basic vector to monitor their promoter activity in vivo in various conditions through a Dual luciferase assay performed on tail lysates at 23 hpf (Fig. 6B-a, E-a). Co-injection of the pGL3-alcama-luc construct with either snai2 MO or pax9 mRNA attenuated the alcama promoter activity, while co-injection with snai2 mRNA or pax9 MO had no significant effect (Fig. 6B-b). Consistent with this, alcama mRNA expression in the tail of pax9 morphants was upregulated at 24 hpf (Fig. 6C), while it was down-regulated in the tail of snai2 morphants (Fig. 6G). We similarly analyzed the effects of *twist1a/twist1b* and *snai2* on the *pdgfra* promoter by co-injection with pGL3-pdgfra-luc. Double knockdown of *twist1a* and *1b* led to a drastic

decrease in *pdgfra* promoter activity in the tail samples, whereas co-injection of twist1b mRNA stimulated it (Fig. 6E-b). In line with this, *pdgfra* mRNA expression in the tail was downregulated in twist1a/1b morphants (Fig. 6F). Snai2 MO also reduced *pdgfra* promoter activity (Fig. 6E-b), and consistently, *pdgfra* expression in the tail of snai2 morphants was significantly reduced (Fig. 6G). Finally, we found that these sclerotomal TFs regulated each other in the tail, as *pax9* knockdown upregulated *snai2* expression but downregulated *twist1b*, while *snai2* knockdown down-regulated *pax9* (Fig. 6D, G). The network of genetic interactions deduced from all these data is depicted in Fig. 6H.

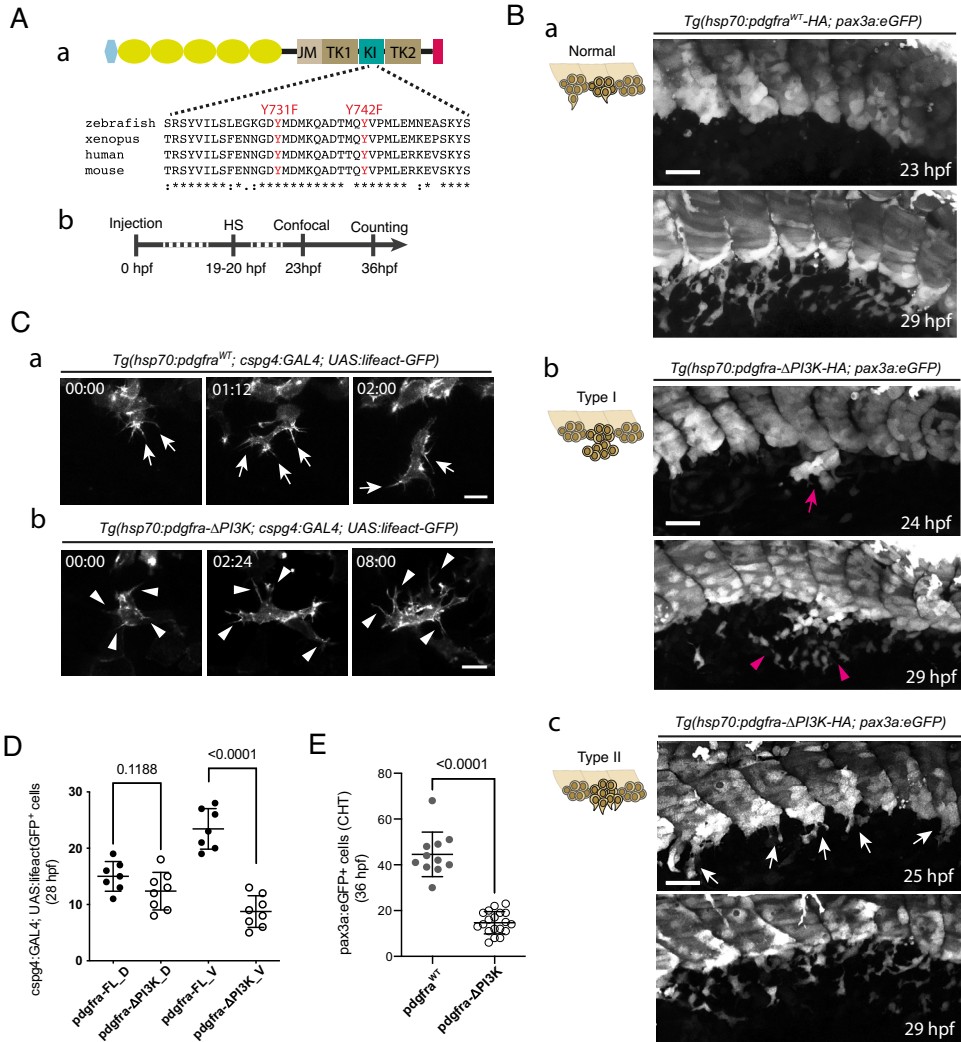

**Fig. 4 | Pdgfr-α influences cluster cohesion and SCP migration behavior through PI3K binding. A** (**a**) Schematic representation of Pdgfra fused to tandem HA-tag (red) at its C-terminus, and amino acid alignment around the site of mutations in the KI (kinase insert) domain. The light blue hexagon and yellow ovals indicate the signal peptide and Ig-like extracellular domains, respectively; JM juxtamembrane, TK1 tyrosine kinase domain 1, TK2 tyrosine kinase domain 2, tyrosine residues involved in PI3K binding when phosphorylated that we mutated into phenylalanines are shown in red. **b** Time course of the experiment; HS, heat-shock. **B** Confocal projections at 23 and 29 hpf of live *Tg(pax3a:eGFP)*embryos injected with *hsp70:pdgfra*$^{WT}$*-HA* (**a**) or *hsp70:pdgfra-ΔPI3K* construct (**b**, **c**) and heat-shocked at 19-20 hpf; *Tg(hsp70:pdgfra-ΔPI3K; pax3a:eGFP)* embryos show defects in VC cohesion (type I, **b**) or SCP emergence and migration (type II, **c**) (see also Supplementary Fig. 4B). Arrow in **b** indicates SCPs overflow out of the cluster before migration initiation. This cell population was dispersed locally after 5 hrs (arrowheads). **c**, Arrows indicate multiple SCPs emerging simultaneously from the cluster; n = 29 (**a**), 7 (**b**), and 13 (**c**) for each condition from 9 independent experiments. Scale bars, 40 μm. **C** Representative frames of a confocal time-lapse sequence starting at 23 hpf of a migrating SCP leader cell in *Tg(cspg4:GAL4; UAS:lifeact-eGFP)* embryo injected with the hsp70:pdgfra-FL (**a**) or hsp70:pdgfra-ΔPI3K (**b**) construct. Time is indicated in hrs and min. Arrows in **a** indicate filopodia projected in the direction of migration; arrowheads in **b** indicate filopodia in all directions. Scale bars, 25 μm. **D** Quantification of *Tg(cspg4:Gal4; UAS:lifeact-eGFP⁺)* SCPs migrating dorsal-wards (D) or ventral-wards (V) in Pdgfra-FL (control) and Pdgfra-ΔPI3K expressing embryos at 28 hpf (n = 7 and 8 embryos for control and hsp70:pdgfra-ΔPI3K, respectively, from 3 independent experiments. mean ± SD; two-tailed Student's t-test. **E** Counting of pax3a:eGFP⁺ SCPs in control and hsp70:pdgfra-ΔPI3K embryos in the CHT at 36 hpf over a 5-somite width (n = 11 and 19 embryos for control and hsp70:pdgfra-ΔPI3K, respectively, from 3 independent experiments). mean ± SD; two-tailed Student's t-test. Source data for **D** and **E** are provided as a Source Data file.

Recently Huang and coworkers found that in addition to the ventral sclerotome found in all vertebrates, zebrafish (and likely other finned vertebrates) possesses a smaller, dorsal sclerotome domain, mirroring the position of the ventral one, i.e. positioned at the dorsal-most border of the somites[15]. This domain is also marked by the classical sclerotome transcription factors, and notably gives rise to the FMCs of the dorsal side of the caudal fin, as well as tenocytes[15] and ISV-associated pericytes and fibroblasts[12], as the ventral sclerotome does. We noted that *alcama* and *pdgfra* are both expressed in this dorsal sclerotome compartment as well as in the ventral one (Figs. 1C, 3F, and 7A, arrowheads) and that Alcama or Pdgfra or PI3K signaling deficiency all caused a strong reduction of FMCs in the dorsal side of the caudal

fin (Figs. 1H, 3F, Supplementary Figs. 3E and 5D, arrowheads), mirroring what we described above for the ventral sclerotome and ventral side of the caudal fin.

## Trunk somites produce alcama and Pdgfr-α dependent stromal cells that foster HSPC emergence

Like the sclerotomal TFs mentioned above, *alcama* and *pdgfra* expression is not restricted to the tail, but also extends to the somites of the trunk (Fig. 7A). Moreover, as in the tail, pax3a:eGFP⁺ or ET37:eGFP⁺ mesenchymal cells are present in the trunk by 26 and 36 hpf, notably just lateral or ventral to the dorsal aorta (DA), and in close contact with it or with the dorsal wall of the underlying axial vein (PCV)

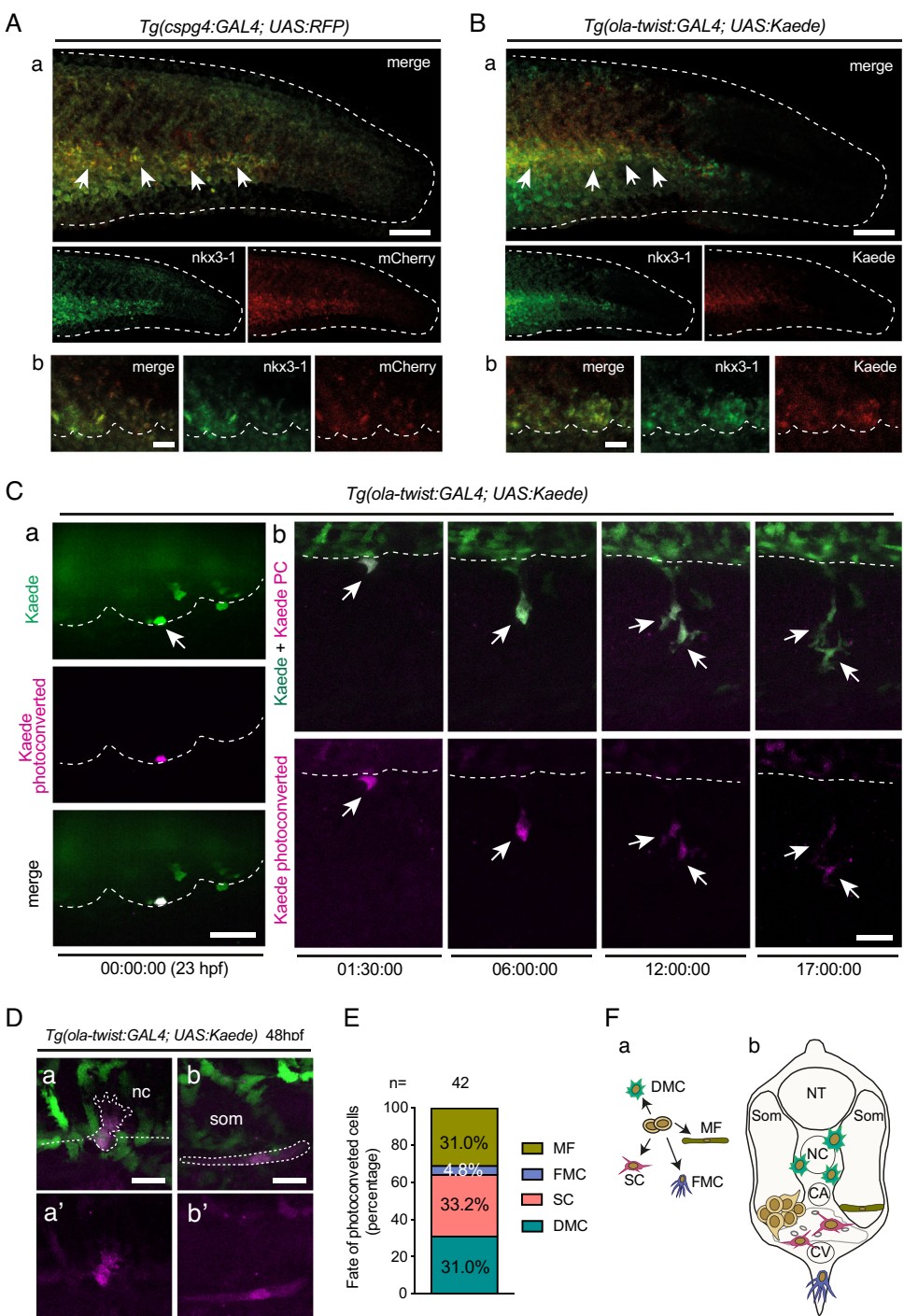

**Fig. 5 | Somite VCs coincide with the sclerotome. A**, **B** *Tg(cspg4:Gal4; UAS:RFP)* embryos (**A**) and *Tg(ola-twist:Gal4; UAS:Kaede)* embryos (**B**) were fixed at 24 hpf and subjected to double fluorescence WISH using probes for *nkx3.1* and *mCherry* or *Kaede* mRNA, respectively. The entire tail area (**a**) and magnified images of the VCs (**b**) are shown; dashed line and arrows in **a** indicate the border of the caudal region and VC cells showing double staining. Dashed lines in **b** delineate the VC ventral borders. Scale bars, 100 μm in (**a**) and 20 μm in (**b**). **C** Confocal images of a live *Tg(ola-twist:Gal4; UAS:Kaede)* embryo. **a** a single Kaede(green)+ cell at the ventral border of an anterior caudal somite was photoconverted (magenta) at 23 hpf. **b** Representative frames from a time-lapse follow-up of the Kaede+ cell photo-converted in (**a**), which underwent mitosis at 10h15 (~33 hpf). Dashed line

delineates the somite ventral border. Scale bars, 20 μm. **D** examples of photo-converted cells that migrated dorsalwards to the notochord (nc) (**a, a′**), or that differentiated in a muscle fiber at the somite ventral border (**b, b′**). Scale bars, 10 μm. **E** Histogram showing the percentage of each cell type differentiated at 48 hpf from single cells photoconverted at 23 hpf. Forty-two cells were analyzed from three independent experiments. **F** (**a**) Schematic representation of the develop-mental fates of photoconverted sclerotome cells; **b** localization of each derivative after migration. DMC dorsal mesenchymal cell, SC stromal cell, MF muscle fiber, FMC fin mesenchymal cell, NC notochord, NT neural tube, CA caudal artery, CV caudal vein, Som somite.

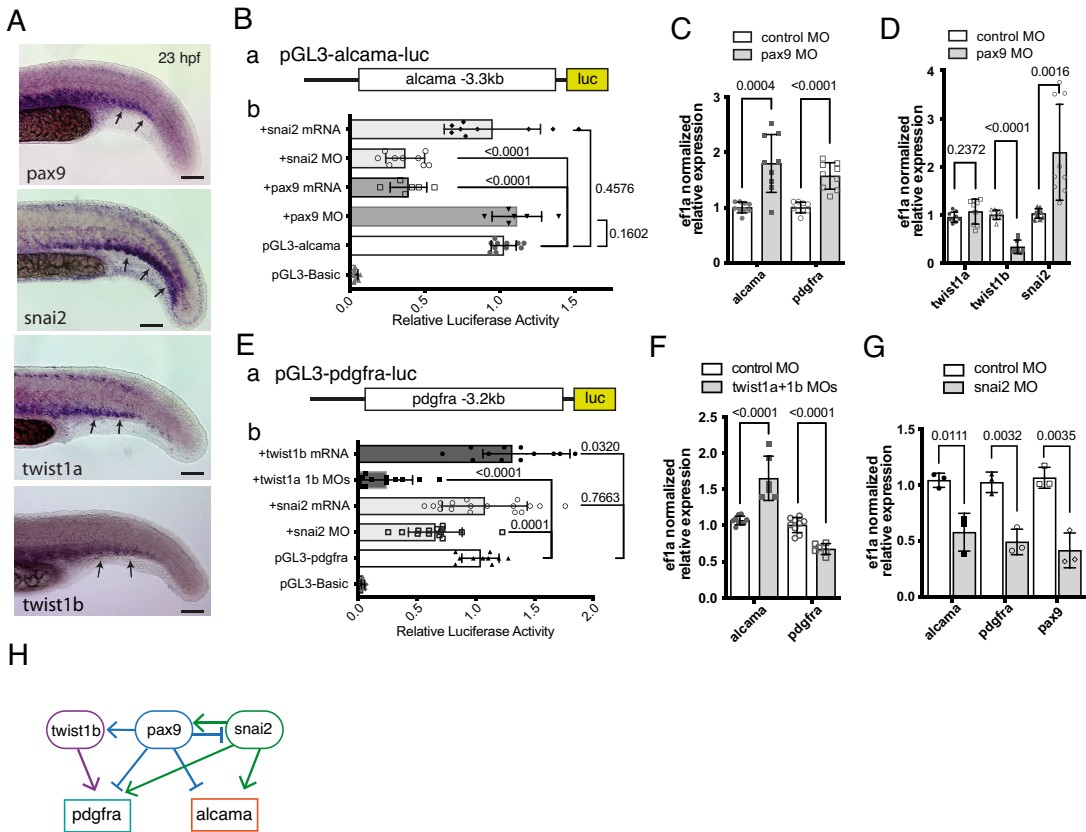

**Fig. 6 | *alcama* and *pdgfra* are regulated by sclerotome-associated TFs. A** WISH for *pax9*, *snail2*, *twist1a*, and *twist1b* at 23 hpf in the tail of WT embryos. Arrows point to expression of these genes in the somitic VCs. **B (a)** a DNA fragment encompassing 3.3 kb upstream of the *alcama* gene transcription start site was inserted upstream of the luciferase (luc) coding region of the pGL3-Basic plasmid, to give pGL3-alcama-luc. **b** In situ luciferase assays at 23 hpf following pGL3-alcama-luc injection at the 1-cell stage either alone or together with the indicated mRNA or MO; each lysate was obtained from 10 to 15 tails (*n* = 9 lysates for each condition, as biological triplicates from three independent experiments) or pax9 (*n* = 6 for each). The pGL3-Basic and pGL3-alcama-luc (*n* = 11 for each) were used as negative and positive controls, respectively (mean ± SD; one-way ANOVA followed by Dunnett's multiple comparison test). **C** qPCR analysis for *alcama* and *pdgfra* in pax9 morphants and control (*n* = 9 pooled tail samples for each, a set of biological triplicates from three independent experiments) at 23 hpf. mean ± SD; two-tailed Student's *t*-test. **D** qPCR analysis for *twist1a*, *twist1b*, and *snai2* in pax9 morphants and control (*n* = 9 pooled tail samples for each, measured examined over three

independent experiments) at 23 hpf. mean ± SD; two-tailed Student's *t*-test. **E (a)** a DNA fragment covering 3.2 kb upstream of the *pdgfra* gene transcription start site was inserted upstream of the luciferase (luc) coding region to give pGL3-pdgfra-luc. **b** Same experimental scheme as in **B-b**, following injection of the indicated constructs and MO or mRNA combinations (*n* = 12, 11, 17, and 14 lysates for twist1b mRNA, twist1a 1b MOs, snai2 mRNA and snai2 MO examined over three independent experiments). mean ± SD; One-way ANOVA followed by Dunnett's multiple comparison test. (**F**) qPCR analysis for *alcama* and *pdgfra* (n = 6 pooled tail samples for each, measured examined over two independent experiments) in twist1a/twist1b double morphants and control at 23 hpf. Mean ± SD; two-tailed Student's *t*-test. **G** qPCR analysis for *alcama*, *pdgfra*, and *pax9* in snai2 morphants and control (*n* = 3) at 23 hpf (mean ± SD; two-tailed Student's *t*-test). **H** Proposed network of TFs acting on *alcama* and *pdgfra* during SCP development; lines ending with arrow or bar represent activation or repression, respectively. Source data for **B**–**G** are provided as a Source Data file.

(Fig. 7B, C-a, D-a; Supplementary Movie 8). Time-lapse confocal imaging showed their dynamics in the sub-aortic space (Supplementary Movie 9). As in the caudal region, we found that Alcama, Pdgfr-a or Pax9 deficiency led to a strong reduction of these pax3a:eGFP⁺ or ET37:eGFP⁺ stromal cells of the trunk (Fig. 7C, D). In addition, these cells were found lateral or ventro-lateral to the aorta, but never just ventral to it (Fig. 7C, D). By 52 hpf, the same defects in their positioning were still observed (Supplementary Figs. 6Aa–d, Ba–d), and their total number in the trunk was even more reduced relative to control embryos than at 36 hpf (Supplementary Figs. 6A-e, B-e).

Totally, 36 hpf is the developmental stage by which definitive HSPCs begin to emerge by EHT from the ventral wall of the aorta—a process that peaks by 48–52 hpf[1,4]. We found that Alcama, Pdgfr-α, or Pax9 deficiency all caused a dramatic reduction in myb⁺ HSPCs in the trunk by 36 hpf (Fig. 7E, F). At the same time point, we could observe the beginning of myb⁺ HSPC colonization of the CHT of control embryos, whereas almost no myb⁺ cell was detected in the CHT of all morphants (Fig. 7E-a). Then, at 52 hpf, only few myb⁺ HSPCs were found in the trunk of all morphants, even less than at 36 hpf, while

myb⁺ cells were numerous in control embryos, and very few myb⁺ HSPCs were detected in the CHT of all morphants (Fig. 7E-b). Thus the severe impact of Alcama or Pdgfr-α deficiency on the development of sclerotome-derived stromal cells both in the trunk (Supplementary Fig. 6C) and in the tail altogether led to profoundly defective definitive hematopoiesis.

## Discussion

Mouse studies have shown the presence of various stromal cell subsets in hematopoietic stem cell niches[8]. However, the developmental process and behavior of these cells, and the molecules involved in cell-cell interactions during niche formation, remain a mystery. In the present study, we have taken advantage of the accessibility to in vivo observations of the zebrafish CHT, the hematopoietic homolog of the fetal liver niche in mammals, to provide new insights into the genesis of stromal cells. We previously found that the stromal cells of the CHT arose from ventral clusters (VCs) within the caudal somites. We report here a new transgenic line, *TgBAC(cspg4:Gal4)*, that specifically labels these VCs and their derivatives, notably the SCPs throughout their

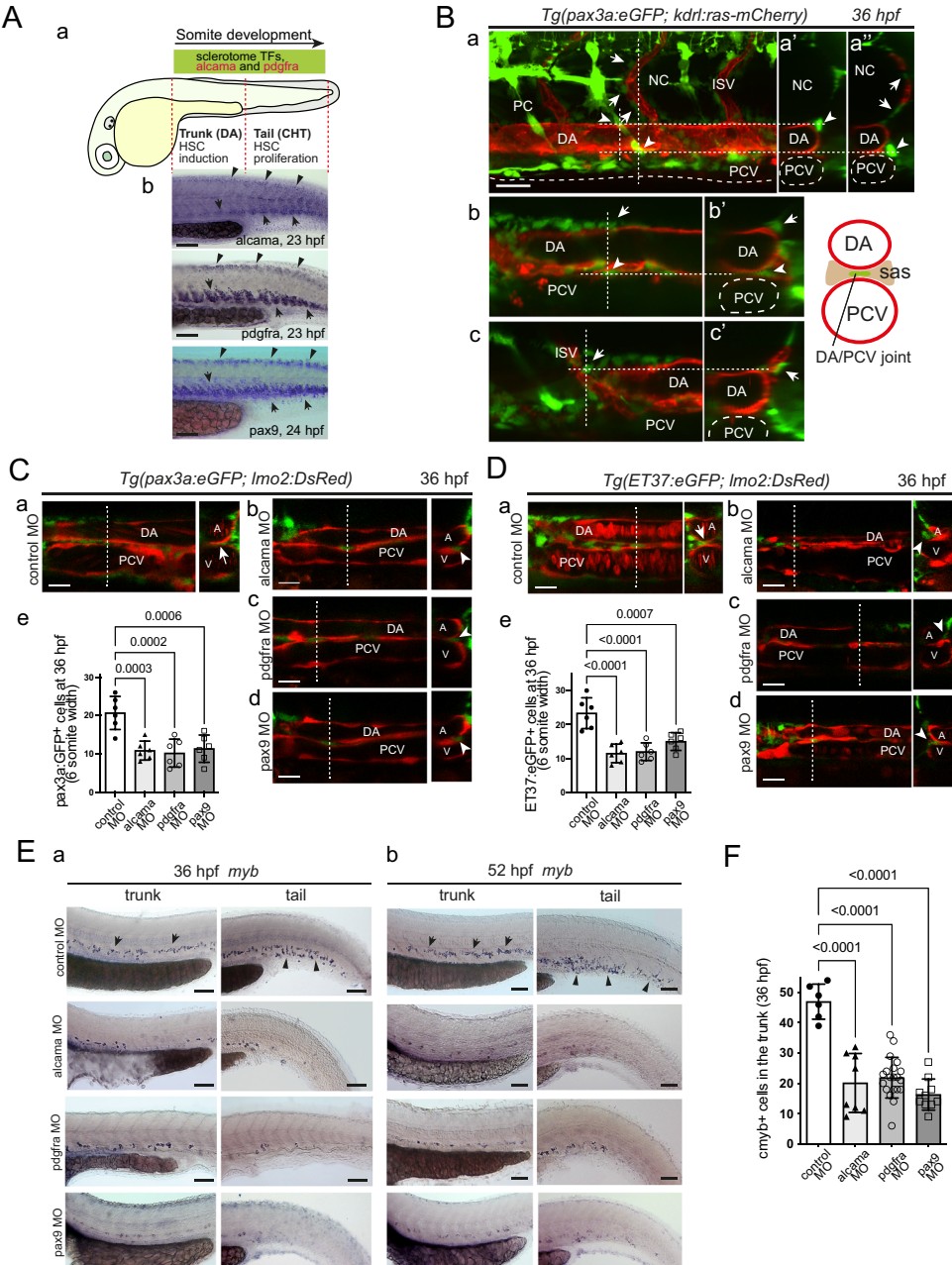

**Fig. 7 | Alcama and Pdgfr-α are also involved in stromal cell development in the trunk, which conditions the development of definitive HSPCs. A** (**a**) Rostro-caudal extent of sclerotomal TFs and *alcama* and *pdgfra* expression by 24 hpf. **b** WISH for *alcama*, *pdgfra*, and *pax9* at 23–24 hpf. Arrows point at their expression in the somite VCs/sclerotome in the trunk and tail. Arrowheads point at their expression in the smaller dorsal sclerotome[15]. **B** Confocal maximum projection (**a**) and single confocal sections (**b**, **c**) of the trunk region of a *Tg(pax3a:eGFP; kdrl:ras-mCherry)* embryo at 36 hpf, with vertical dashed lines showing the position of the optical transverse sections shown on the right (**a'**, **a''**, **b'**, **c'**). Arrowheads point at GFP+ stromal cells located at the DA/notochord or DA/PCV junction, arrows point at ISV-associated mesenchymal cells, known to originate from the dorsal-wards migration of ventral sclerotome cells[12,15]. Scale bar, 20 μm. **C, D** Confocal sections of the trunk region of *Tg(pax3a:eGFP; lmo2:DsRed)* (**C**) and *Tg(ET37:eGFP; lmo2:DsRed)* (**D**) embryos injected with control (**a**), alcama (**b**), pdgfra (**c**), or pax9 (**d**) MOs. Dashed lines indicate the position of the corresponding transverse section shown on the right. Arrows and arrowheads point at stromal cells located at the DA/PCV

joint or somewhat lateral to it, respectively. **e** Quantification of pax3a:eGFP+ (**C**) or ET37:eGFP+ (**D**) cells in the sub-aortic space of 36 hpf live embryos injected with control, alcama, pdgfra or pax9 MO (mean ± SD; *n* = 6 embryos for each group. One-way ANOVA followed by Dunnett's multiple comparison test). Counting was done over a 6-somite width. Scale bars, 20 μm. **E** WISH for *myb* at 36 (**a**) and 52 (**b**) hpf in embryos injected with control, alcama, pdgfra, or pax9 MO. Arrows and arrowheads point at myb+ signals in the trunk (arrows) and tail (arrowheads) regions, respectively. **a** Representative images of *n* = 6, 8, 20, and 10 injected embryos for control, alcama, pdgfra, and pax9 MO, respectively, from two independent experiments. **b** Representative images of *n* = 6 for each condition, from two independent experiments. Scale bars, 100 μm. **F** Quantification of myb+ cells at 36 hpf as shown in (**E-a**) (mean ± SD; *n* = 6 embryos for control, 8 for alcama, 20 for pdgfra, and 10 for pax9 MO-injected embryos. One-way ANOVA followed by Dunnett's multiple comparison test. DA or A dorsal aorta, PCV or V posterior cardinal vein, ISV intersomitic vessel, sas sub-aortic space. Source data for **C-e**, **D-e**, and **F** are provided as a Source Data file.

development, and at least two other types of mesenchymal cells—dorsal-wards migrating cells (DMCs) and fin mesenchymal cells (FMCs)[35].

VC cells specifically express sclerotome marker genes such as *nkx3.1*, *pax9*, *twist1a/b*, and *snai2*. We found via double WISH and in vivo tracking of sclerotomal cells from the caudal somites that the VCs indeed coincide with the sclerotome. Historically the sclerotome has been defined as the ventro-medial layer of the early, still epithelial somites that become mesenchymal cells via an EMT, and then migrate to encircle the notochord and differentiate into chondro- and osteo-genic derivatives to form the vertebrae. Later it was found to also give rise to other migrating mesenchymal cells that would become mural cells (pericytes and vascular smooth muscle cells-vSMCs) ensheathing the dorsal artery and other vessels, both in amniotes[11] and more recently in zebrafish[12,13,14] as well as tenocytes (tendon cells) attaching somite muscle fibers to the bones[15], and perivascular fibroblasts[12]—a cell population that would likely later also give rise to adipocytes[43]. To these sclerotomal derivatives we are now adding, for the caudal region, the stromal cells of the CHT, and the mesenchymal cells of the ventral side of the caudal fin (FMCs, that will become fibroblasts in the adult fin[35]), as similarly found by Ma et al.[15] for the dorsal side of the caudal fin. Considering that the cell population originating from the caudal somites that differentiate into osteoblasts in adult fish[44] may also be included, cells from the VCs/sclerotome may be considered as multi-potent mesenchymal progenitors, with a range of fates wider than the threefold (chondro-, osteo-, and adipo-genic) potential usually assayed in vitro to assert the Mesenchymal Stem Cell (MSC) capability of cells isolated from various neonatal or adult tissues in mammals[45]. Strikingly, the three transmembrane molecules that we found here to be more specifically expressed in the VCs and their derivatives, Alcama, Pdgfr-α, and Cspg4 (also known as NG2), have all been used in mammals as markers of bone marrow and/or fetal liver mesenchymal stromal cells to purify these cells, that were then often shown to be endowed with MSC capability in vitro[8,23,32,33,46]. This further suggests to us that like the stromal cells of the zebrafish CHT, stromal cells of mammalian hematopoietic niches in the trunk and limbs may have a sclerotomal origin and that Alcama and Pdgfr-α will no longer be seen as mere markers of these cells, but as essential for their ontogeny.

It will be important to determine when and how the "commitment" process[47], which corresponds to the first branching point in the differentiation of multipotent mesenchymal cells into various cell lineages, may take place within the sclerotome. Our tracking of caudal sclerotome-derived cells presently suggests that their fate within the VCs may already be restricted by somite maturation stage S8. The development of genetic tools to mark the sclerotome earlier will be needed to conclude on this point.

We further found that Alcama-, Pdgfr-α-, and Pax9-dependent, hence sclerotome-derived pax3a:eGFP+/ET37:EGFP+ cells are present not only in the tail but also in the trunk, as mesenchymal cells associated with the DA and underlying axial vein (PCV), and that they foster the emergence of HSPCs from the DA. Nguyen et al.[48] previously found somite-derived pax3a:GFP+ cells that fostered HSPC emergence, and they interpreted them as partly endothelial cells (of the DA and other vessels) and partly "vascular associated cells" ("VACs"). In our own analysis, we found all pax3a:GFP+ cells to be VACs, part of them in very close apposition to the DA or PCV endothelium, highly reminiscent of the stromal mesenchymal cells that we had previously imaged at that same position and developmental stages by electron microscopy[6]. These cells most likely include precursors of the mural cells (vSMCs and pericytes), known to derive from the sclerotome[12,14]. Studies by Traver and colleagues[49,50] found that HSPC ontogeny required somitic Wnt16-induced Dlc/Dld signaling to angioblasts by 15–17 hpf during their migration towards the midline along the ventral surface of somites, where the early sclerotome lies. Therefore the sclerotome may possibly foster HSPC emergence at two successive stages, first

through such direct contact with and signaling to migrate DA precursors, and later through the interaction of sclerotome-derived mesenchymal cells with the DA hemogenic endothelium.

Thus the sclerotome successively gives rise to mesenchymal cells involved in the first two main stages of HSPC development and function. Trunk somites first contribute mesenchymal stromal cells that foster the emergence of HSPCs from the DA floor, then also likely facilitate their intravasation into the circulation through the dorsal wall of the underlying PCV. We indeed previously found by electron microscopy stromal mesenchymal cells bridging gaps between endothelial cells of the PCV dorsal wall through which we had shown that aorta-derived HSPCs enter blood circulation[6]. Then the later formed caudal somites contribute stromal cells that are key components of the niche where the aorta-derived HSPCs will first settle, expand and undergo multi-lineage differentiation. We found that Alcama and Pdgfr-α expression in the sclerotomal compartments of the trunk and tail are both necessary to give rise to these two sets of mesenchymal stromal cells that are successively essential for the development of definitive hematopoiesis. In addition, they appear similarly involved in the generation of mesenchymal cells of the caudal fin from both the ventral and dorsal sclerotome compartments.

We identified the earliest morphological individualization of the caudal sclerotome as a distinct ventral cell cluster appearing at somite maturation stage S5 (in agreement with Morin-Kensicki et al.[9] for the trunk somites) and encompassing the whole mediolateral extent of the somite before these cells become mesenchymal and migrate both dorsal- and ventral-wards. Interestingly, Naganathan et al.[51] recently found that the antero-posterior length of freshly formed somites is somewhat variable and that over the next 1–2 h (i.e., until somite stage S4), they undergo a mechanical adjustment of this A–P length that is facilitated by somite surface tension, which requires the somite to be fully packed within an uninterrupted basal lamina. It, therefore, makes sense that cell migrations out of the somite, which require breaking the basal lamina, can be "allowed" to begin only once this mechanical rearrangement of the somite has been completed, i.e. from somite stage S4/S5.

Morphological individualization of VCs by somite stage S5 as a 3D 'rosetta' with a small central lumen appears concomitant with the onset of expression at the same place of the TF genes *twist1a/b* and *pax9*, which may thus trigger the EMT, and of *pdgfra*. Since *alcama* and *snai2* transcripts are already detected at the somite ventral and dorsal borders by stage S1/S2 (Figs. 1C and 6A), our data of Fig. 6B–G suggest that Snai2 may be sufficient to activate *alcama* transcription at stages S2–S4, whereas Twist1b may be additionally required for the activation of *pdgfra* transcription as well as for the EMT by stage S5 (Supplementary Fig. 7).

Unlike alcama transcripts, the Alcama protein first appears in the center of the cluster by stage S5 and then gradually spreads to all interfaces among cluster cells. Then as these cells emigrate from the cluster, they do so as strings of cells still enriched in Alcama at their interfaces. Suppression of Alcama function reduced the number of emigrating SCPs, their contacts among each other, and their migratory capacity. The latter defect correlated with a less efficient polarization of the cell and its F-actin dynamics relative to the ventral direction of migration. When ventral cluster cells overexpressed a mutant Alcama deleted of its two N-terminal Ig domains (that mediate homophilic cell-cell adhesion), they started displaying extreme filopodial dynamics while still within the clusters. This may be favored by the lack of Alcama-mediated cell-cell contacts, while the still present short ICD of this overexpressed ΔN-Alcama form may trigger more F-actin dynamics generating filopodia. Conversely, ventral cluster cells over-expressing a mutant Alcama form in which the ICD is no longer able to bind PDZ domain-containing actin-binding proteins such as syntenin-1[26–28] showed a lack of filopodial dynamics, more extended cell-cell contacts, and an even more complete lack of apparent cell polarity.

This phenotype could be related to the fact that when the link of transmembrane Alcama molecules to the cytoskeleton is suppressed, they are freer to cluster within the membrane, which in turn reinforces Alcama-mediated cell–cell contacts[52].

Suppression of Pdgfr-α function affected the emergence and ventral-wards migration of SCPs at least as strongly as Alcama deficiency, whereas neither affected the dorsal-wards migration of DMCs. Our finding that both Alcama and Pdgfr-α deficiencies led to a strong decrease in pERK signaling in the ventro-caudal region suggests that the intracellular signaling pathways downstream of Alcama and Pdgfr-α overlap within the SCPs. In addition, we found PI3K activation to be an obligatory step downstream of Pdgfr-α activation for the development of these cells. Interestingly, the dorsal-wards migration of sclerotome cells was also affected in embryos treated with a PI3Kα,γ inhibitor, indicating that even though distinct pathways are likely involved in the dorsal and ventral migrations of sclerotome derivatives, both use PI3K as a necessary downstream effector (Supplementary Fig. 7).

Our new *Tg(cspg4:Gal4)* line, which highlights these cells from their sclerotomal origin onwards, will be a precious tool to discern, notably through single-cell transcriptome analysis, whether cellular heterogeneities prefiguring different fates are already present within the somitic VCs and how different the stromal cells fostering HSPC emergence from the aorta are from those that then nurture HSPC expansion and multi-lineage differentiation in the CHT niche. Elucidation of the molecular mechanisms of cell commitment within the multipotent sclerotome should contribute to the field of regenerative medicine by facilitating the induction and manipulation of rare MSC populations from adult tissues.

## Methods

### Zebrafish
Fish were maintained in our zebrafish facility at Institut Pasteur. Embryos were obtained through natural crosses, raised at 28 °C in embryo water [Volvic® water containing 0.28 mg/ml Methylene Blue (M-4159; Sigma) and 0.03 mg/ml 1-phenyl-2-thiourea (P-7629; Sigma)], and staged according to Kimmel et al.[53] For this study, we used the previously described transgenic lines *Tg(ET37:EGFP)*[54], *Tg(pax3a:EGFP)*[il50 55], *Tg(UAS:Lifeact-GFP)*[mu271 36], *Tg(7xTCF-Xla.Siam:nlsmCherry)*[56], *Tg(kdrl:ras-mCherry)*[57] *Tg(lmo2:DsRed)*[58] and *Tg(ola-twist:Gal4; UAS:Kaede)*[35].

### BAC recombineering and transgenesis
The tol2 and BAC recombineering vectors were kindly provided by K. Kawakami (National Institute of Genetics, Mishima). A BAC DNA containing the *cspg4* gene (DKEY-105I23) was purchased from Source BioScience. The *TgBAC(cspg4:Gal4)* line was firstly generated by recombining a iTol2-amp cassette to the BAC vector[59], then a second recombineering was performed targeting a cassette containing GAL4 (pGAL4FF-FRT-Kan-FRT) flanked by 50 bp arms homologous to the region around the translation start of the cspg4 gene. Primers used for the 1st and 2nd recombineering steps are listed in Supplementary Table 1). Recombination was performed using RedET methodology (K001, GeneBridges) as described in a previous report[60]. BAC DNAs were prepared using NucleoBond BAC 100 (740579, Marchery-Nagel). *Tg(UAS:RFP)* embryos were injected with 1 nl of mixed solution of 100 ng/μl BAC DNA and 50 ng/μl Tol2 mRNA at the 1-cell stage.

### In situ hybridization, immunostaining, and Western blotting
Whole-mount RNA in situ hybridization (WISH) was performed according to Thisse (https://wiki.zfin.org/display/prot/Thisse+Lab+-+In+Situ+Hybridization+Protocol+-+2010+update). Riboprobes were synthesized by reverse transcription from PCR fragments amplified from cDNA synthesized from the tails of WT embryos as templates, or from linearized plasmids. The probes used in this study are indicated in Supplementary Tables 2a and 2b.

Double WISH was performed with the following modifications to the above protocol. Riboprobes were synthesized using Digoxigenin-11-UTP or Fluorescein-12-UTP (Roche) then signal detection was performed by horseradish peroxidase (POD)-based method followed by tyramide-based signal amplification (TSA)[61]. To increase the sensitivity of probe detection containing Fluorescein-12-UTP, the probe was first reacted with an anti-Fluorescein-R-phycoerythrin antibody (Invitrogen, A21250, 1/100) followed by an anti-rabbit-POD secondary antibody (Sigma, A0545, 1/200).

Whole-mount fluorescent immunostaining was carried out as described[5]. The list of primary and secondary antibodies and concentrations used is shown in Supplementary Table 3.

For the Western blotting, tails at 24 hpf were solubilized on ice in lysis buffer (50 mM Tris-HCl, 150 mM NaCl, 2.5 mM EDTA, 1% Triton X100, 1X final protease inhibitors cocktail, pH 8) at a ratio of 20 tails/20 μl. After 30 min solubilization on ice, samples were centrifuged at 4 °C for 10 min at 13,000 rpm. Supernatants were transferred to new tubes and aliquots of the samples were treated for protein determination using BCA reagents (Sigma) with BSA standards ranging from 62.5 to 250 mg/ml. Samples were then adjusted for loading 67.5 μg per well of total protein. After SDS-PAGE (12% acrylamide gels), proteins were transferred onto nitrocellulose (Protran, 0.45 μm) using a Genie Blotter system (Idea Scientific Company). Membranes were then incubated for 60 min in blocking buffer (Tris Buffered Saline (TBS) complemented with 0.1% tween-20 (TBST) and 5% w/v low-fat milk). Anti-Alcama monoclonal antibodies (Zn-8, Developmental Studies Hybridoma Bank) were diluted in blocking buffer (1/2000) and incubated with membranes overnight, at 4 °C. After extensive washing in TBST, membranes were incubated with peroxidase-coupled anti-mouse antibodies diluted in TBST for 2 h at room temperature. Complexes were revealed via chemiluminescence using ECL Prime western blotting detection reagents (ThermoFisher Scientific) and the Bio-Rad Chemidoc MP imaging system.

### Construction of mutant forms of Alcama and Pdgfra, transient transgenesis, and heat-shock treatment
**UAS:alcama-ΔN-eGFP, UAS:alcama-ΔPDZ-eGFP and UAS:alcama-FL-eGFP.** We first cloned the alcama ORF to utilize it as a template for the following PCRs (details of cloning are indicated below). For *alcama-ΔN* construct, a DNA fragment missing to encode amino acids 25–238 corresponding to the two amino-terminal Ig-like domains of Alcama was amplified. For the *alcama-ΔPDZ* construct, 3 amino acid substitutions were introduced in a putative PDZ domain binding motif at positions 533–539 (KTRQGSW->**MV**RQGS**G**) by site-directed mutagenesis. Primers used for the mutagenesis are listed in Supplementary Table 4. Control alcama-FL (full length) and mutated alcama fragments were cloned into the Nco I site of UAS:eGFP vector in frame with eGFP using Gibson assembly (#E5510, NEB).

**hsp70:pdgfra-ΔPI3K-HA and hsp70:pdgfra<sup>WT</sup>-HA.** A zebrafish pdgfra-ΔPI3K was designed based precisely on a human dominant-negative pdgfra, with the substitutions Y706F (Y731F in humans) and Y717F (Y742F)[29]. Fragments containing amino acid substitutions were amplified by PCR using the clone containing full-length WT pdgfra (kindly provided by J. Eberhart, University of Texas at Austin) as a template. A 3 kb fragment covering the ORF was amplified in three fragments and the point mutations were introduced in the appropriate fragments using primers listed in Supplementary Table 4. A tandem HA-tag sequence was added in the frame at the C-terminus of pdgfra. Fragments encoding control or mutant pdgfra were cloned into hsp70 vector using Gibson assembly.

Plasmid DNA for each construct was co-injected with capped mRNA coding for the Tol2 transposase into 1-cell stage embryos at the amount of 250 and 25 pg, respectively. Embryos were heat-shocked by

placing them in pre-warmed Volvic water for 30 min at 39 °C then transferred to 28 °C until in vivo observation.

## RNA/cDNA synthesis, plasmid construction, and injection

Total RNA was extracted with the TRIzol reagent (15596026, Invitrogen) from tails of anesthetized embryos at 23 hpf. Tail total RNA was reverse transcribed into cDNA using a Superscript IV reverse transcriptase (18091050, Invitrogen). For *alcama, pdgfaa, pdgfab, pdgfbb,* and *twist1b*, ORFs were amplified by PCR using tail cDNA as a template for the subcloning into the TOPO vector and then reinserted into pExpress1 vector using Gibson assembly. ORFs of *pax9* and *snai2* were amplified by PCR from tail cDNA then cloned directly into pExpress1 vector. Primers used for the cloning are listed in Supplementary Table 5. mRNAs were synthesized using mMessage mMachine transcription kit (AM1344, Ambion). Capped mRNAs were injected into 1-cell stage embryos at the amount of 100 pg.

## Morpholinos and qPCR

Morpholino oligonucleotides (Gene Tools) were injected (0.5–1 nl) into the 1-cell-stage embryo at the amount specified; alcama MO[20] (2 ng), pdgfra MO[40] (4 ng), pax9 MO[62,63] (6 ng), snai2 MO[63,64] (8 ng), and twist1a and twist1b MOs[65] (2 ng each). MO sequences are shown in Supplementary Table 6. For quantitative real-time PCR (qPCR), total RNA was extracted from 3 independent groups of 40 tails dissected from embryos at 23 hpf to synthesize template cDNAs. The primers used for qPCR are shown in Supplementary Table 7. All qPCR experiments were performed with measurements taken from 3 technical replicates. Fold changes in gene expression were calculated using the $2^{-\Delta\Delta CT}$ method and normalized to ef1a.

## Promoter cloning and Dual-Luciferase assay

3.3 and 3.2 kb promoter regions of *alcama* and *pdgfra* genes were cloned from BAC DNA (*alcama*: RP71-78P1, BACPAC Resources; *pdgfra*: DKEY-97C6, Source BioScience) and cloned into the pGL3 basic vector (E1751, Promega) linearized with Xho I and Hind III (pGL-alcama-luc and pGL-pdgfra-luc). Primers used for promoter cloning are listed in Supplementary Table 8. To assess *alcama* and *pdgfra* promoter activities in response to various TFs, WT embryos were injected at the 1-cell stage with luciferase vectors (pGL-alcama-luc or pGL-pdgfra-luc), control (Renilla) expression vector (pRL-TK; E2241, Promega), and a specified amount of Morpholino or appropriate mRNA encoding TF (co-injection condition is listed in Supplementary Table 9. Ten to fifteen tails were dissected from the injected embryos at 23–24 hpf and lysed with passive lysis buffer (E1941, Promega) for subsequent Dual-Luciferase assay following the manufacturer's instructions (E1910, Promega).

## Drug treatments

Pharmacological inhibitors were all solubilized in DMSO (Sigma) and appropriate DMSO controls (0.2%) were used for all experiments. Embryos treated with AS605240 (2 μM; S1410, Selleckchem), LY294002 (20 μM; S1105, Selleckchem), and CAL101 (10 μM; S2226, Selleckchem) from 19-20 hpf embryos then live-images were taken with drugs in agarose and embryo water supplemented with tricaine. For WISH, embryos were treated with a drug from 19 to 20 hpf until the stage of interest, then fixed in 4% methanol-free formaldehyde overnight.

## Generation of alcama and pdgfra-crispants

Transient CRISPR-Cas9 targeting of the alcama and pdgfra genes was performed by using Alt-R CRISPR-Cas9 system (Integrated DNA Technologies / IDT). The gRNA target sites (shown in Supplementary Table 10) were evaluated for the off-target sites by using the sgRNA design tool (IDT). Alt-R S.p. Cas9 V3 enzyme (IDT) and equimolar amount of crRNA:tracrRNA duplexes (IDT) were mixed and incubated

to form RNP complexes before injection. The mixed solution was diluted with Cas9 buffer (NEB) to adjust the concentration of Cas9 and sgRNA working solution to 12 and 18 μM, and 1 nl was injected into the cytoplasm of one-cell stage *Tg(pax3a:eGFP; kdrl:ras-mCherry)* embryos. Injected F0 crispant embryos were individually screened by genomic PCR using primers that recognize outer sites of the region sandwiched by a pair of gRNA (for alcama) or primers designed at intron2/exon3 of the pdgfra gene. For the cDNA analysis of pdgfra-crispants, total RNA was extracted from 100 to 120 pooled embryos and RT-PCR was performed using primers designed at exon2/exon4 to check the resulting splicing patterns. The primers used are shown in Supplementary Table 10.

## Live imaging, image processing, and quantification

VE-DIC (video-enhanced differential interference contrast) microscopy was performed using a Polyvar2 microscope (Reichert) as described previously[66]. Confocal fluorescence microscopy was performed on Leica SPE and SP8 set-ups and Andor spinning disk set-ups as described previously[4,7]. Image processing was basically performed using Fiji[67]. Measurements of surface area and volume on migrating SCPs were performed using IMARIS software (Bitplane). Cell tracking was performed using a Manual tracking plugin (Fiji) and the obtained data were (1) plotted with a Python script developed by S. Rigaud (Image Analysis Hub (IAH) of Institut Pasteur) or (2) imported into the Chemotaxis and Migration Tool plugin[68] (Fiji). For the analysis of filopodial dynamics, Filopodyan plugin[69] was used in Fiji. First, we obtained a time-lapse movie (6 min intervals, 23–38 hpf) of the migrating lifeact-GFP+ SCPs injected with control or alcama MO (7 leader cells were analyzed for each condition). We extracted 7–10 timepoints from the time that the leader cell left its followers and measured the orientation of the filopodia for each timepoint. The cut-off threshold was set to 3 μm. The direction of SCP migration was obtained from the positional information at the first and last time-points. The angle of filopodia relative to the direction of SCP migration was calculated with a Python script developed by D. Ershov (IAH, Institut Pasteur). Photoshop and Illustrator (Adobe) were used to combine two or more confocal images to display a wide view (joints are indicated by dotted lines in each figure).

## Single-cell photoconversion and confocal imaging

Photoconversion was achieved using an Andor (Oxford Instruments) spinning disk confocal system (CSU-W1 Dual camera with 50 μm disk pattern and single laser input (445/488/561/642 nm), LD Quad 405/ 488/561/640 and triple 445/561/640 dichroic mirrors), equipped with a Leica DMi8 fluorescence inverted microscope, a Digital Mirror Device (DMD-Mosaic 3 (Andor)), and CMOS cameras (Orca Flash 4.0 V2 + (Hamamatsu)). *Tg(ola-twist:Gal4; UAS:Kaede)* embryos were kept in the dark at 28 °C until 23 hpf then mounted in 1% low-melting agarose. Photoconversion of the Kaede in single cells at the ventral side of caudal somites S8-S11 at 23 hpf was performed at 408 nm, using a 63x water immersion objective (HC PL APO 63x/1.20 W CORR CS2), a LEDs light source (CoolLED pE-4000 16 wavelength LED fluorescence system), and the DMD-Mosaic-3 with the support of the MetaMorph software. After photoconversion, embryos were kept mounted at 28 °C for 1 h and subsequently time-lapse imaged at 6 min intervals using a Leica SP8 confocal microscope as described above.

## Statistics and reproducibility

Statistical analysis was performed using GraphPad Prism 9. For data that followed a normal distribution analyzed by the Shapiro–Wilk test, statistical significance was assessed by a two-tailed Student's *t*-test, while Mann–Whitney test was introduced for two groups with unequal variances. One-way ANOVA was performed to analyze the differences among multiple groups, followed by Tukey's or Dunnett's multiple comparison test. Data are shown as mean or median ± SD. For the

calculation of exact p values below $p < 0.0001$ for $t$-tests, the $t$-score and degrees of freedom (DF) values were added to the source data. For ANOVA, SS (sums of squares), MS (mean square), DF and F distribution were also added to the appropriate source data. For confocal analysis for Tg embryos, expression patterns were confirmed in at least 6 independent animals. All WISH patterns were confirmed in at least 6 independent animals. In vivo observations of non-treated wild-type embryos were performed at least five times, and WISH and IHC were basically performed at least three times unless stated otherwise. For qPCR and luciferase assays, results were collected with three biological replicates.

## Reporting summary

Further information on research design is available in the Nature Portfolio Reporting Summary linked to this article.

## Data availability

Raw imaging files are available upon request due to their large size. Source data are provided in this paper.

## Code availability

All software and plug-ins used in this study are either freely or commercially available.

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

## Acknowledgements

We thank our fish facility team for zebrafish care, K. Kawakami (National Institute of Genetics, Mishima) for providing us with the BAC recombineering plasmids, T.J. Carney (IMCB, Singapore) for Tg(ola-twist:Gal4; UAS:Kaede) fish, R. Patient (University of Oxford) for the hsp70-4:pG1 plasmid, J. Eberhart (University of Texas at Austin) for the pdgfra:pCR4 plasmid, B. Appel (University of Colorado) for the cspg4:pJC53.2 plasmid, and A. Kudo (Tokyo Institute of Technology, Nagatsuta) for the pax9:pPICT3 plasmid. We also thank S. Rigaud, D. Ershov, and J.-Y. Tinevez (IAH of Institut Pasteur) for the implementation of Python scripts for filopodia analysis. We thank L. Tchon (Fondation de l'AP-HP) for working with us on our initial trials on the pax9 project. We thank A.-L. Touret (Institut Pasteur) for her contribution to setting up the Western blot experiments, and K. McElreavey, D. Houzelstein, and C. Eozenou (Institut Pasteur) for their help with Luciferase assays. This work was supported by Institut Pasteur, CNRS, and by grants to P.H. from the Fondation pour la Recherche Médicale (#DEQ20160334881), the Fondation ARC pour la Recherche sur le Cancer, and the Agence Nationale de la Recherche Laboratoire d'Excellence Revive (Investissement d'Avenir; ANR-10-LABX-73).

## Author contributions

E.M. conceived the project, designed, performed, and analyzed experiments, and wrote the paper. C.V. conducted experiments and fish care. A.S. conducted experiments and assisted with helpful discussions. P.H. ensured the funding of the project, analyzed, discussed, and suggested experiments, and wrote the paper.

## Competing interests

The authors declare no competing interests.
