## [Peer Review File · Nature Communications]

Alcam-a and Pdgfr- α are essential for the development of sclerotome-derived stromal cells that support hematopoiesis.REVIEWER COMMENTS

Reviewer #1 (Remarks to the Author):

The manuscript entitled "Alcam-a and Pdgfr-a are essential for the development of sclerotome derived stromal cells that support hematopoiesis in vivo." by Murayama et al. presents an impressive analysis of the emergence and development of stromal cell progenitors (SCPs) in the zebrafish caudal hematopoietic tissue (CHT). The authors previously showed that SCPs emerge from the somites and form the mesenchymal component of the CHT required for hematopoietic stem and progenitor cell (HSPC) colonization of the niche. In their current manuscript they extend that work by looking at the molecular signals that induce and guide SCP emergence and migration.

The homophilic adhesion protein Alcam-a and the receptor tyrosine kinase Pdgfr-a, are found to be expressed in the SCPs coincident with their emergence from the somites. Alcam-a in particular is shown to localize to adhesive contacts within cellular rosettes in the ventral somites from which the SCPs bud off. Morpholino (MO) inhibition of either gene disrupts SCP budding and migration into the nascent CHT. This in turn leads to a reduction in the number of SCPs as well the CHT fibroblastic reticular cells that are derived from them. In order to target SCPs more specifically the authors identified the gene *cspg4*, homologous to the mammalian gene *CSPG4/NG2*, as a conserved marker for stromal cells and created fish expressing *cspg4:Gal4*. Expression of *cspg4:Gal4/UAS:lifect-GFP* revealed F-actin-rich filopodia predominantly on the dorsal side of migrating SCPs while no directional bias was found in Alcam-a morphants. Furthermore clonal expression of UAS:Alcam-a or UAS:Pdgfr-a mutants in SCPs revealed this to be a cell autonomous defect and surprisingly only disrupted ventral but not dorsal migration, suggesting divergent signals for each migratory path. Despite this difference both migration paths remained dependent on PI3K and ERK signaling. The authors then investigated whether mesenchymal transcription factors (TFs) influenced gene expression from the *alcam-a* and *pdgfr-a* promoters. Together with detailed qPCR measurements of these genes in TF morphants provided evidence for a gene regulatory network driving expression of *alcam-a* and *pdgfr-a* in SCPs. Finally, the authors show that in addition to SCPs role in forming the CHT they also induce formation of HSPCs from the ventral aorta. SCPs are located in direct contact with the hemogenic endothelium and *alcam-a*, *pdgfr-a*, and *pax9* morphants all have reduced aortic SCPs as well as a reduction in HSPCs budding.

The authors have shown a clear role for the conserved stromal cell membrane proteins Alcam-a and Pdgfr-a in SCP formation and migration. Interestingly they have discovered that these proteins are required for the ventral migration of SCPs into the CHT but not for dorsal migration towards the notochord, revealing divergent signals required for distinct SCP populations. Additionally, they prove that SCPs are the mesenchymal cells responsible for inducing HSPC emergence from the aorta and clarify previous reports. This work is both novel and relevant for developmental biologists and hematopoietic researchers. Their *cspg4* based SCP expression system is also extremely useful and will provide a key tool for genetically targeting and investigating these cells in the future. The methodology utilized throughout is impressive with excellent developmental biology, cell biology, and molecular biology all supporting their conclusions. We believe this manuscript is well worthy of publication in Nature Communications.

Major Points:

1. The identification of the SCP cells as sclerotomal in origin is not well supported. The primary method of distinguishing the mesodermal population is the SCP migratory path to the notochord and the expression of the TFs Pax9, Snai2, Twist1a and Twist1b. While their location and Pax9 expression are appropriate sclerotomal indicators the remaining TFs are considered general mesenchymal markers and may label other non-sclerotomal mesenchymal populations such as the dermatome. The expression of more sclerotome specific genes such as *nkx3.2* in the SCPs would bolster this point.

2. A better discussion of the evidence for a sclerotomal origin of SCPs is also needed. Notably the recent papers from Peng Huang's laboratory that describe the development and migration of sclerotomal cells in zebrafish should be addressed directly in the text.

3. The expression pattern of *cspg:Gal4* should be examined in more detail. While there is major overlap of this marker with other SCP labels such as *ET37:GFP* and *Pax9:GFP* there are also notable areas without co-expression even within the CHT region. Remarks on the penetrance of this expression system within the presumptive SCPs as shown in Fig1 would be appropriate. Additionally, images of the whole body revealing expression of *cspg:Gal4* compared to either of these markers would be prudent to reveal all tissues encompassed.

4. TF network description (Fig 5) and the authors model for how this TF network functions developmentally (Fig. S6) should be expanded upon in the text. The timing and expression pattern of the TFs in the somites postulated in Fig S6 should also be supported or discussed as part of a speculative model.

Minor Points:

- Movie S2 is confusing and should be re-rendered with better time stamps. Ideally annotations should be used to highlight relevant areas during the many pauses, time jumps, and rotations/reorientations of the movie as these manipulations make it hard to follow the development of the SCPs.
- Filopodia angle measures in Fig 2B-C should define leader cell (are leader cells ventral migrating specific or are dorsal migrating cells included, and if so what is the proportion of each migratory population).
- VE-DIC should be defined in the text and/or figure legend.
- Fig 4-B and Fig S4-C have split control and experimental data, ideally these should be displayed together.
- Figure call outs for Fig S5 and Fig S6 should be more detailed and pertinent

Reviewer #2 (Remarks to the Author):

Murayama et al extend their previous studies about the stromal cells that contribute to the hematopoietic niche in the zebrafish caudal hematopoietic tissue display additional markers associated with mesenchymal stem cell identity, that the previously-identified ventral somitic origin of these cells can be assigned the identity of the sclerotome, and that the two genes that feature in these studies (*Alcama* and *Pdgfra*) are contribute to processes in the caudal hematopoietic tissue niche and hematopoietic stem/progenitor cell (HSPC) induction on the ventral aorta. Its major contribution is the description and functional focus of these two genes.

The manuscript is tightly written and well prepared. Overall, I have very little to comment on about the experimental work itself, which I considered to be rigorous, logically developed and well presented.

MAJOR COMMENTS

1. The introduction is primarily a narrative of work done by these investigators. The work would better be place in perspective if it provided some context from wider work in the field. While this occurs to

some extent in the discussion, the report would be improved by a more inclusive initial picture. A recent informative review is Tani et al (2020) PMID:32788657 – the understandings in this make it evident that it is unsurprising in the field that the sclerotome is involved either in HSPC or mesenchymal stem cell (MSC) development. While I was reviewing this report, a relevant paper was published (Lummertz da Rocha et al, Nat Cell Biol PMID:35414020) in which the sclerotome features as a contributor to HSPC development. (Of course, the authors could not be expected to have cited this work, but it indicates that the field in which this work sits is broader than then picture cast in this manuscript's introduction).

2. Fig 3D. The enlarged images do not contain all the signal in the boxed regions. The far left panels appear to be MIPs; are the enlarged images possibly single imaging planes? In the case of the pdgfra MO panels, I could recognise no correlation between the enlarged image and the boxed area in the panel to the left (unlike the two scenarios above, where the a visual correlation was recognisable)

3. Discussion paragraph 1. The priority claims in this paragraph invite scrutiny. The conceptual link between paraxial mesoderm and sclerotome and MSCs is not new (e.g PMID:33573345), although this manuscript does report experiments that actually support this concept. The use of the two markers in this paper invites the question, what is a MSC? It was unclear to me whether the expression of some markers associated with MSCs was sufficient to assign multipotent MSC identity – the text seems to cautiously avoid doing so (abstract line 7-8; introduction page 4 line 12-13), and the possibility of there already being some restricted potency is acknowledged (lines 24-26). Suggest some rewriting to better nuance the positioning of this work in the field.

OTHER COMMENTS

1. Much of page 5 is a describes series of initial explorations. Make it more focused on what was directly helpful.

2. Statistics. Where more than two groups are being compared, use an ANOVA rather than multiple t-tests to minimize the risk of type 1 error (e.g. Fig. 3E; Fig 5B, E (I presume these are t-tests, it is not stated here as it is elsewhere); Fig. 6C, D, F; Fig S3D).

3. Scale bars. Fig 1F - provide scale bar for magnified images. Fig 3 – no images have scale bars. Fig S1A – provide scale bars.

4. Replication. From p28 line 20-21 (but representative of many such statements) "mean±SD; n=11 for control, n=19 for hsp70:pdgfra-DPI3K, from 3 independent experiments." Do the statements in this format mean that the 11 control datapoints are POOLED from 3 experiments? I ask for confirmation because it means the individual experiments had only 3-4 embryos per group, which seems unexpectedly small.

5. Fig 6. In many of the images, seeing the vertical dashed white lines indicating the level of the optical section were not visible at magnifications below 200% (I initially thought they were missing). Suggest making them more obvious.

6. Page 18 Line 18. "drastically reduced" is reduced to levels about 40% of control-MO (panel b); remove "drastically".

7. Fig. S1F. The Western. blot needs an equal loading control for the reduced/negative signal in the MO lanes to be meaningful. This made me look for the Western method and it is not provided.

8. Fig 2 and Movie 4. At first I was puzzled from the internal figure labels of Fig 2 by why all the red cells (expressing UAS:RFP) were not also green (from the UAS:alcama...GFP constructs). It was only on reading the Movie 4 legend that I realised that the experimental design of injecting the latter as a construct would result in its mosaic expression. Other readers will likely be helped by this aspect of the experimental design being explained explicitly in the text and main figure (by inserting "resulting in mosaic expression" or similar).

Reviewer #3 (Remarks to the Author):

The study by Murayama et al is a further characterisation of a potential stromal cell population that the authors have previously described arise from the ventral part of the caudal somites in zebrafish

through an epithelial mesenchymal transition. As described in their previously published work these stromal cell progenitors migrate to form a component of the caudal hematopoietic tissue niche, which acts to coordinate hematopoietic stem cell maturation. In this new work the authors modestly extend these previous observations to

“more closely characterized” the process of the emergence of these cells from the somites. The authors contend that these cells emerge from the sclerotome based on marker-based studies and morpholino knockdown and use morpholino knockdown to implicate specific genes in their formation.

Major Points.

1. Although the exact source of the stromal cells of the CHT within the somites is an interesting point for those interested in the HSC field, it is a finding obviously derivative of the original interesting observation of the authors. The developmental biology of the formation of the specific somite population that give rises to it, although interesting, in my opinion is likely better suited to a DB focussed journal.

2. The loss of function analyses exclusively relies on morpholinos. Given the ease of using CRISPR technologies this is some what surprising. Given the extensive literature on the pitfalls of using these tools in isolation this seriously detracts from the impact of the manuscript. I do acknowledge the use of mutant forms of these genes over expressed in vivo but true genetic loss of function studies were needed

3. Alcama and Pdgfr-a, are widely expressed markers, Pdgfr-a for instances marks fibroblasts in a wide variety of tissues. I was not sure then what these markers were actually marking? This probably needed to be better defined and the fate of these somite cells (not the expression of different transgenes and gene expression but defined lineage analyses) determined.

4. Loss of the sclerotome is known to result in loss of signals that support cell type specification more broadly in the somite. To better establish the sclerotomal origin a true fate mapping strategy using a sclerotome marking CRE line (or photoconvertible fate map of the same) would be needed to be certain of a sclerotomal origin of these cells.

Minor comments .

1. Some of the referencing and attribution of previous studies is sloppy.

For instance, two important papers from the Traver group are not referenced.

Kobayashi I, Kobayashi-Sun J, Kim AD, Pouget C, Fujita N, Suda T, Traver D. Jam1a-Jam2a interactions regulate haematopoietic stem cell fate through Notch signalling Nature. 2014 Aug 21;512(7514):319-23.

Clements WK, Kim AD, Ong KG, Moore JC, Lawson ND, Traver D. A somitic Wnt16/Notch pathway specifies haematopoietic stem cells. Nature. 2011 Jun 8;474(7350):220-4.

These are important papers that have primacy in the field for defining the way somitic signals regulate HSC formation and are not discussed in any detail.

Secondly the authors have mis-interpreted the results of another important study.

Nguyen PD, Hollway GE, Sonntag C, Miles LB, Hall TE, Berger S, Fernandez KJ, Gurevich DB, Cole NJ, Alaei S, Ramialison M, Sutherland RL, Polo JM, Lieschke GJ, Currie PD Haematopoietic stem cell induction by somite-derived endothelial cells controlled by meox1. Nature. 2014 Aug 21;512(7514):314-8.

The authors contend two points about this paper: That this work shows that somite cells that colonise the dorsal aorta derive from the dermomyotome-derived cells and that their work contradicts these findings as their stromal cell, they contend, derive from the sclerotome. In actual fact Nguyen et al specifically show that these cells do not derive from cells that give rise the External cell layer the functional equivalent of the amniote dermomyotome. They instead derive from a somite compartment that expresses unique (non-sclerotome) markers that they show by lineage analyses to be distinct to External cell layer progenitors. In fact they form in a mutually exclusive manner. Nguyen et al term this compartment the “endotome” as they show by lineage analyses that they give rise to endothelial cells in the dorsal aorta. Clearly, based on the data in the two analyses, it is obvious that these two studies describe two different cells types (in fact the Nguyen et al study examines the DA and does not examine CHT dynamics in any detail).

2. There is clear overreach on the interpretation of the authors results. This is best typified by the

statement at the end of the abstract:

"Thus the sclerotome contributes essential stromal cells for each of the key steps of developmental hematopoiesis, and likely is the embryological origin of most if not all mesenchymal stem/stromal cell found in non-cephalic tissues."

Unfortunately there is simply no evidence in the submission to support this statement. No fate mapping at all is reported in this study let alone in the majority of non-cephalic tissues.

Response to the Reviews (in blue)

Reviewer #1 (Remarks to the Author):

The manuscript entitled “Alcam-a and Pdgfr-a are essential for the development of sclerotome derived stromal cells that support hematopoiesis in vivo.” by Murayama et al. presents an impressive analysis of the emergence and development of stromal cell progenitors (SCPs) in the zebrafish caudal hematopoietic tissue (CHT). The authors previously showed that SCPs emerge from the somites and form the mesenchymal component of the CHT required for hematopoietic stem and progenitor cell (HSPC) colonization of the niche. In their current manuscript they extend that work by looking at the molecular signals that induce and guide SCP emergence and migration.

The homophilic adhesion protein Alcam-a and the receptor tyrosine kinase Pdgfr-a, are found to be expressed in the SCPs coincident with their emergence from the somites. Alcam-a in particular is shown to localize to adhesive contacts within cellular rosettes in the ventral somites from which the SCPs bud off. Morpholino (MO) inhibition of either gene disrupts SCP budding and migration into the nascent CHT. This in turn leads to a reduction in the number of SCPs as well the CHT fibroblastic reticular cells that are derived from them. In order to target SCPs more specifically the authors identified the gene *cspg4*, homologous to the mammalian gene *CSPG4/NG2*, as a conserved marker for stromal cells and created fish expressing *cspg4:Gal4*. Expression of *cspg:Gal4/UAS:lifect-GFP* revealed F-actin-rich filopodia predominantly on the dorsal side of migrating SCPs while no directional bias was found in Alcam-a morphants. Furthermore clonal expression of UAS:Alcam-a or UAS:Pdgfr-a mutants in

SCPs revealed this to be a cell autonomous defect and surprisingly only disrupted ventral but not dorsal migration, suggesting divergent signals for each migratory path. Despite this difference both migration paths remained dependent on PI3K and ERK signaling. The authors then investigated whether mesenchymal transcription factors (TFs) influenced gene expression from the *alcam-a* and *pdgfr-a* promoters. Together with detailed qPCR measurements of these genes in TF morphants provided evidence for a gene regulatory network driving expression of *alcam-a* and *pdgfr-a* in SCPs. Finally, the authors show that in addition to SCPs role in forming the CHT they also induce formation of HSPCs from the ventral aorta. SCPs are located in direct contact with the hemogenic endothelium and *alcam-a*, *pdgfr-a*, and *pax9* morphants all have reduced aortic SCPs as well as a reduction in HSPCs budding.

The authors have shown a clear role for the conserved stromal cell membrane proteins Alcam-a and Pdgfr-a in SCP formation and migration. Interestingly they have discovered that these proteins are required for the ventral migration of SCPs into the CHT but not for dorsal migration towards the notochord, revealing divergent signals required for distinct SCP populations. Additionally, they prove that SCPS are the mesenchymal cells responsible for inducing HSPC emergence from the aorta and clarify previous reports. This work is both novel and relevant for developmental biologists and hematopoietic researchers. Their *cspg4* based SCP expression system is also extremely useful and will provide a key tool for genetically targeting and investigating these cells in the future. The methodology utilized throughout is impressive with excellent developmental biology, cell biology, and molecular biology all supporting their conclusions. We believe this manuscript is well worthy of publication in Nature Communications.

Major Points:

1. The identification of the SCP cells as sclerotomal in origin is not well supported. The

primary method of distinguishing the mesodermal population is the SCP migratory path to the notochord and the expression of the TFs Pax9, Snai2, Twist1a and Twist1b. While their location and Pax9 expression are appropriate sclerotomal indicators the remaining TFs are considered general mesenchymal markers and may label other non-sclerotomal mesenchymal populations such as the dermatome. The expression of more sclerotome specific genes such as *nkx3.2* in the SCPs would bolster this point.

- Even though *Snai2* and *Twist1a/b* are indeed involved in various EMT events in development and regeneration, within the somites of zebrafish embryos *twist* expression was found to mark specifically the sclerotome (Morin-Kensicki et al., 1997; Stickney et al., 2000); so does *snai2* (Bickers et al. 2018). Neither of these “markers” was allocated to the dermatome (Stellabotte & Devoto, 2007). In the more recent study by Peng Huang’s laboratory (Ma et al., 2018), the genes documented to mark the entire sclerotome compartment were first *nkx3.1* then *pax9*, *twist1b* and others (whereas *nkx3.2* only later marked a subset of sclerotome cells that had migrated to the notochord). Therefore, since we had already documented *pax9* expression, we have now documented by double in situ hybridization that in the caudal somites by 24 hpf, the expression of *nkx3.1* fully coincides with that of our novel transgenic marker *Tg(cspg4:gal4; UAS:RFP)* (**new Fig. 5A**). Furthermore, in the somite maturation process, *nkx3.1* and *cspg4:gal4* expression both become detectable at somite maturation stage S4/S5.

2. A better discussion of the evidence for a sclerotomal origin of SCPs is also needed. Notably the recent papers from Peng Huang’s laboratory that describe the development and migration of sclerotomal cells in zebrafish should be addressed directly in the text.

- We do agree. In fact our manuscript initially included a long paragraph explicitly connecting our study to the two papers from P. Huang’s team (Ma et al. 2018; Rajan et al. 2020), which we ultimately removed in order to comply with the manuscript size limits of the journal, while still mentioning these two references so that they were at least kept in our reference list. We have now re-introduced this paragraph in the revised manuscript (**p. 13**), and mentioned again their work in the Discussion (**p. 15**). Notably, the Huang lab’s clarification that in fish, unlike in amniotes, the sclerotome arises from not only a ventral domain of the somites but also from a symmetrically positioned smaller domain at the dorsal border of the somites is fully consistent with the expression domains of the *alcama* and *pdgfra* genes that we have observed.

We have also reinforced our evidence of the sclerotomal origin of SCPs by using the *Tg(ola-twist1:Gal4; UAS:Kaede)* line, as this and a *Tg(ola-twist1:eGFP)* line have been used to label the sclerotome in zebrafish (Lee et al. 2013, Stratman et al. 2017; Ma et al. 2018). We first checked that in such transgenic embryos, Kaede expression coincided with *nk3.1* expressing cells (**new Fig. 5B**); then we used photoconversion of Kaede in single cells of the somite ventral clusters, and followed up the fate of the photoconverted cells (**new Fig. 5C-F**), which confirmed our previous data on the multiple fates of the somite ventral cluster cells.

3. The expression pattern of *cspg:Gal4* should be examined in more detail. While there is major overlap of this marker with other SCP labels such as ET37:GFP and Pax9:GFP there are also notable areas without co-expression even within the CHT region. Remarks on the penetrance of this expression system within the presumptive SCPs as shown in Fig1 would be appropriate.

- Indeed, in live imaging, RFP expression from the *Tg(cspg4:Gal4; UAS:RFP)* appears mosaic among ET37:GFP-positive SCPs that colonize the CHT, particularly up to 36 hpf (less so thereafter). Such mosaicism is a known feature of expression using the Gal4/UAS system in zebrafish; we have now mentioned that in the text (**pp. 5, 7/8, and Fig.2-legend**) Interestingly, when we increased the detection sensitivity of RFP expression (in the double in situ hybridization expression experiment of the new **Fig. 5A**, wherein we used a double

amplification of RFP signal detection), we did detect RFP expression in all *nkx2.1+* cells (as mentioned above in our response to Point 1), which means that the mosaicism of RFP expression seen in vivo does not reflect an all-or-none expression, but merely a lower level of RFP expression in a fraction of the SCPs.

Additionally, images of the whole body revealing expression of *cspg:Gal4* compared to either of these markers would be prudent to reveal all tissues encompassed.

- We now provide such high-resolution images of whole double transgenic *Tg(cspg4:Gal4; UAS:RFP; ET37:GFP)* and *Tg(cspg4:Gal4; UAS:RFP; Pax3a:GFP)* embryos at 24 and 48 hpf in **Supplementary Fig. 1D,E**.

4. TF network description (Fig 5) and the authors model for how this TF network functions developmentally (Fig. S6) should be expanded upon in the text. The timing and expression pattern of the TFs in the somites postulated in Fig S6 should also be supported or discussed as part of a speculative model.

- We have added the following passage in the Discussion (**p.17**):

“Morphological individualization of VCs by somite stage S5 as a 3D 'rosetta' with a small central lumen appears concomitant with the onset of expression at the same place of the TF genes *twist1a/b* and *pax9*, which may thus trigger the EMT, and of *pdgfra*. Since *alcama* and *snai2* transcripts are already detected at the somite ventral and dorsal borders by stage S1/S2 (Figs. 1C and 5A), our data of Fig. 5B-G suggest that *Snai2* may be sufficient to activate *alcama* transcription at stages S2-S4, whereas *Twist1b* may be additionally required for the activation of *pdgfra* transcription as well as for the EMT by stage S5 (Supplementary Fig. 7).”

Minor Points:

- Movie S2 is confusing and should be re-rendered with better time stamps. Ideally annotations should be used to highlight relevant areas during the many pauses, time jumps, and rotations/reorientations of the movie as these manipulations make it hard to follow the development of the SCPs.

- We had introduced these rotations so as to make visible the 3-dimensional position of the highlighted cells relative to somite structure. We have now added annotations at key points in the movie, stages in hpf, as well as a more clear timer through the time-lapse sequence, and have simplified the rotations by suppressing the 360° rotation that was interrupting the time-lapse sequence.

- Filopodia angle measures in Fig 2B-C should define leader cell (are leader cells ventral migrating specific or are dorsal migrating cells included, and if so what is the proportion of each migratory population).

- As mentioned in the text, we analyzed only the SCP leader cells, i.e. cells migrating toward the ventral side.

- VE-DIC should be defined in the text and/or figure legend.

- Indeed. This has now been corrected (we had previously defined it only in the Methods section).

- Fig 4-B and Fig S4-C have split control and experimental data, ideally these should be displayed together.

- We have now inserted the previous Fig. S4C into the main Fig. 4 (Fig. 4B-a).

• Figure call outs for Fig S5 and Fig S6 should be more detailed and pertinent

- Indeed. We have now done so (pp. 13,14,17,18; note that the previous Figs. S5 and S6 have become S6 and S7).

Reviewer #2 (Remarks to the Author):

Murayama et al extend their previous studies about the stromal cells that contribute to the hematopoietic niche in the zebrafish caudal hematopoietic tissue display additional markers associated with mesenchymal stem cell identity, that the previously-identified ventral somitic origin of these cells can be assigned the identity of the sclerotome, and that the two genes that feature in these studies (*Alcama* and *Pdgfra*) are contribute to processes in the caudal hematopoietic tissue niche and hematopoietic stem/progenitor cell (HSPC) induction on the ventral aorta. Its major contribution is the description and functional focus of these two genes.

The manuscript is tightly written and well prepared. Overall, I have very little to comment on about the experimental work itself, which I considered to be rigorous, logically developed and well presented.

MAJOR COMMENTS

1. The introduction is primarily a narrative of work done by these investigators. The work would better be place in perspective if it provided some context from wider work in the field. While this occurs to some extent in the discussion, the report would be improved by a more inclusive initial picture.

- We wrote the Introduction as the actual history/background of what brought forth the study that we are presenting. As it is the development of this study that led us to the sclerotome/MSC connection, we then put its results in the perspective of the sclerotome and MSC literature in the Discussion. Nevertheless, we have now added more context about the sclerotome in the Introduction of the revised manuscript.

A recent informative review is Tani et al (2020) PMID:32788657 – the understandings in this make it evident that it is unsurprising in the field that the sclerotome is involved either in HSPC or mesenchymal stem cell (MSC) development. While I was reviewing this report, a relevant paper was published (Lummertz da Rocha et al, Nat Cell Biol PMID:35414020) in which the sclerotome features as a contributor to HSPC development. (Of course, the authors could not be expected to have cited this work, but it indicates that the field in which this work sits is broader than then picture cast in this manuscript's introduction).

- There is no mention of MSCs nor of HSPCs in this review by Tani et al (2020). Even in their later review that you mention below (Tani et al. 2021; PMID:33573345), they do not link MSCs to the somites – quite the opposite in fact (see our response to Point 3).

- The paper by Lummertz da Rocha et al. 2022 is a mere data base of single-cell transcriptome data from the mouse AGM area, leading to clustering them in cell groups then putatively ascribed to the cell types known or suspected to be present in the AGM area, with suggested molecular interactions merely based on the transcriptome data. So by construction, all cell groups present in the AGM, including the sclerotome, are present in their figures. The only mention of the sclerotome in the whole text of this article is: "Our

intercellular communication network of the AGM revealed that cellular states defined as endothelial cell 1 (EC1), HE, mesonephros, urogenital ridges (UGRs) and somites (sclerotome and dermomyotome) as well as the mesenchymal subset mesenchymal stem and progenitor cell 1 (MSPC1) interact with EC1, HE and T1preHSC populations through a higher diversity of ligand–receptor pairs compared with the neural and muscle lineages (Fig. 2b).“ - i.e. a most general and predictable statement... The only experimental data presented to validate the transcriptomics/informatics analysis are unrelated to the sclerotome.

2. Fig 3D. The enlarged images do not contain all the signal in the boxed regions. The far left panels appear to be MIPs; are the enlarged images possibly single imaging planes?

- Indeed. We have now mentioned this in the legend. And the right-most image is an orthogonal projection (“optical transverse section”), at the position indicated by a vertical line in the middle panel (which we have now made more obvious, as it was too thin to be discerned at medium magnification).

In the case of the *pdgfra* MO panels, I could recognise no correlation between the enlarged image and the boxed area in the panel to the left (unlike the two scenarios above, where the a visual correlation was recognisable)

- Thank you very much for having noticed this ! We checked and realized that indeed the MIP on the left side was a wrong one (from another embryo). We have now replaced it by the one matching the single plane and orthogonal projections shown on the right.

3. Discussion paragraph 1. The priority claims in this paragraph invite scrutiny. The conceptual link between paraxial mesoderm and sclerotome and MSCs is not new (e.g PMID:33573345), although this manuscript does report experiments that actually support this concept.

- The cited review by Tani et al. 2021 (PMID:33573345) does not make a conceptual link between paraxial mesoderm/sclerotome and MSCs - quite the contrary in fact. As explained in the second paragraph of their Introduction and Fig. 1, they present two contrasting strategies by which osteogenic tissue might possibly be produced in vitro: 1) by using Pluripotent Stem Cells (iPSCs or ESCs) to try and obtain paraxial mesoderm-like cells in a dish, and from there sclerotome and its osteogenic derivatives. 2) By using MSCs (they write: “In addition to the PSC-derived cells, other cell sources can be utilized for skeletal regeneration. These cells are mainly isolated from bone tissue. MSCs are the most conventional type and have been widely studied”). In contrast, our work supports the prospect that developmentally MSCs may originate in the still undifferentiated sclerotome, thereby unifying the two above views.

The use of the two markers in this paper

- three markers actually (*alcama*, *pdgfra* and *cspg4*)

invites the question, what is a MSC? It was unclear to me whether the expression of some markers associated with MSCs was sufficient to assign multipotent MSC identity – the text seems to cautiously avoid doing so (abstract line 7-8; introduction page 4 line 12-13), and the possibility of there already being some restricted potency is acknowledged (lines 24-26). Suggest some rewriting to better nuance the positioning of this work in the field.

- This comment has been most useful to us and we are grateful for it. It indeed led us to dig the literature more deeply to trace the evolution of the MSC concept, and thus to better appreciate the controversies around it (see e.g. our new ref. 45). We rewrote this part of the Discussion with this new awareness (pp.15-16 and 18/19).

OTHER COMMENTS

1. Much of page 5 is a describes series of initial explorations. Make it more focused on what was directly helpful.

- The first part introduces a notion that is foundational for this study: that because each somite develops with a 30 min time lag relative to its caudal neighbor, any dynamic analysis of their development needs to number them not with the conventional numbering more familiar to most biologists, but with the dynamic S1-Sn numbering used e.g. in the somite clock studies, which expresses their maturation stage at a given developmental time. This is what allowed us to get to a more precise model of somite VC/ early sclerotome development. The second part introduces the transgenic lines that we have used to live image the development of SCPs, and this is also essential. We have shortened it a bit, as much as we reasonably could.

2. Statistics. Where more than two groups are being compared, use an ANOVA rather than multiple t-tests to minimize the risk of type 1 error (e.g. Fig. 3E; Fig 5B, E (I presume these are t-tests, it is not stated here as it is elsewhere); Fig. 6C, D, F; Fig S3D).

- Agreed. We have now done so.

3. Scale bars. Fig 1F - provide scale bar for magnified images. Fig 3 – no images have scale bars. Fig S1A – provide scale bars.

- Yes, done.

4. Replication. From p28 line 20-21 (but representative of many such statements) “mean±SD; n=11 for control, n=19 for hsp70:pdgfra-DPI3K, from 3 independent experiments.” Do the statements in this format mean that the 11 control datapoints are POOLED from 3 experiments? I ask for confirmation because it means the individual experiments had only 3-4 embryos per group, which seems unexpectedly small.

- These experiments involved the confocal time-lapse imaging of live embryos, so only 3-4 embryos could be observed at a time.

5. Fig 6. In many of the images, seeing the vertical dashed white lines indicating the level of the optical section were not visible at magnifications below 200% (I initially thought they were missing). Suggest making them more obvious.

- Indeed; we have now done so.

6. Page 18 Line 18. “drastically reduced” is reduced to levels about 40% of control-MO (panel b); remove “drastically”.

- Page 18 Line 18 was in the M&M. We suppose you rather meant p.8, line 19 (“drastically decreased”) ? OK, we replaced “drastically” by “reduced to 36% of control level”.

7. Fig. S1F. The Western. blot needs an equal loading control for the reduced/negative signal in the MO lanes to be meaningful. This made me look for the Western method and it is not provided.

- We have now added the detailed methodology for Western blotting in the Methods section. Each sample was subjected to protein quantification before electrophoresis, and equal amounts (67.5 µg) per sample were loaded for control MO and alcama MO injected embryos.

8. Fig 2 and Movie 4. At first I was puzzled from the internal figure labels of Fig 2 by why all the red cells (expressing UAS:RFP) were not also green (from the UAS:alcama...GFP constructs). It was only on reading the Movie 4 legend that I realised that the experimental design of injecting the latter as a construct would result in its mosaic expression. Other readers will likely be helped by this aspect of the experimental design being explained explicitly in the text and main figure (by inserting “resulting in mosaic expression” or similar).

- Agreed. We have now specified this in the text. The mosaic character is due both to the fact that the alcama variant construct was injected in the observed embryos, and to the fact that the GAL4:UAS system typically produces some mosaicism in expression (it is also the case for the UAS:RFP transgene, as we have now also mentioned, p.5; see our response to Reviewer #1's Point 3).

Reviewer #3 (Remarks to the Author):

The study by Murayama et al is a further characterisation of a potential stromal cell population that the authors have previously described arise from the ventral part of the caudal somites in zebrafish through an epithelial mesenchymal transition. As described in their previously published work these stromal cell progenitors migrate to form a component of the caudal hematopoietic tissue niche, which acts to coordinate hematopoietic stem cell maturation. In this new work the authors modestly extend these previous observations to “more closely characterized” the process of the emergence of these cells from the somites. The authors contend that these cells emerge from the sclerotome based on marker-based studies and morpholino knockdown and use morpholino knockdown to implicate specific genes in their formation.

Major Points.

1. Although the exact source of the stromal cells of the CHT within the somites is an interesting point for those interested in the HSC field, it is a finding obviously derivative of the original interesting observation of the authors. The developmental biology of the formation of the specific somite population that give rises to it, although interesting, in my opinion is likely better suited to a DB focussed journal.

- We obviously don't think so. It is also brings a quite interesting prospect to the scientists working on mammalian hematopoietic niches and MSCs as it strongly suggests that Alcama and Pdgfra, which they have been widely using as markers to purify mesenchymal stromal cells notably of the bone marrow, without ascribing any functional significance to these “markers”, had actually been essential for the developmental ontogeny of these cells, and that their MSC capability in vitro likely is a remnant of their sclerotomal origin. In addition, and importantly, our study also shows that in the trunk, similar alcama- and pdgfra-dependent stromal cells develop, and are essential there for the very emergence of HSPCs from the aorta, before their homing to the CHT – thus that such cells are actually essential both for the emergence of HSPCs from the aorta in the trunk and later for nurturing their proliferation and multi-lineage differentiation in the tail.

2. The loss of function analyses exclusively relies on morpholinos. Given the ease of using CRISPR technologies this is some what surprising. Given the extensive literature on the pitfalls of using these tools in isolation this seriously detracts from the impact of the

manuscript. I do acknowledge the use of mutant forms of these genes over expressed in vivo but true genetic loss of function studies were needed.

- We have now performed crispr based analysis for both *alcama* and *pdgfra*, and they confirmed the phenotypes previously obtained with the morpholinos and overexpression of mutant forms of the genes (see **new Supplementary Fig. 5**).

3. *Alcama* and *Pdgfr-a*, are widely expressed markers, *Pdgfr-a* for instances marks fibroblasts in a wide variety of tissues. I was not sure then what these markers were actually marking? This probably needed to be better defined and the fate of these somite cells (not the expression of different transgenes and gene expression but defined lineage analyses) determined.

- Our study was concerned with *Alcama*, *Pdgfra* and *cspg4* expression in the caudal region of the zebrafish embryo (and in its last part, p.13-14, with the similar pattern observed in the trunk, which led us to discover similar *alcama*- and *pdgfra*-dependent stromal cells there). We have clearly defined that *Alcama*, *Pdgfra* and *Cspg4* expression in the caudal region mark the somite ventral cell clusters (VCs) and their migrating progeny. Moreover our taking in account the different maturation stages of successive somites at any time point (each somite being 30-min younger than its immediate rostral neighbor) has allowed us to precisely determine at which stage each somite VC started to detectably express each of the genes that we have considered (*alcama*, *pdgfra*, *cspg4*, and sclerotomal TFs), and to correlate this temporal sequence with the live morphological appearance of the VCs. Our high-resolution immunofluorescence imaging of *Alcama* protein in this framework allowed us to discover that it first appeared in the center of the VC at the precise stage when the VC became morphologically visible (stage S5), and then gradually spread to all cell interfaces within the cluster during the next 1.5 hr (stages S6-S8), after which we could detect it at the interface of the migrating VC cells. To our knowledge, such a spatio-temporal resolution has never been reached in previous studies of sclerotome development, and it has been key to the dynamic model of VC/sclerotome development that we have presented (Fig. S8). We have now further confirmed that the VCs constitute the first morphological appearance of the sclerotome through double in situ hybridization, which showed that (*cspg4:Gal4; UAS:RFP*) expression fully coincided with expression of the earliest known sclerotome marker, *nkx3.1* (**new Fig. 5A**), and actually appeared at the same somite maturation stage.

4. Loss of the sclerotome is known to result in loss of signals that support cell type specification more broadly in the somite. To better establish the sclerotomal origin a true fate mapping strategy using a sclerotome marking CRE line (or photoconvertible fate map of the same) would be needed to be certain of a sclerotomal origin of these cells.

We have now used local photo-conversion of *Kaede* in *Tg(ola-twist1:Gal4; UAS:Kaede)* embryos, in which *ola-twist1* was previously reported to direct *Kaede* expression to the sclerotome. We first checked that in such transgenic embryos, *Kaede* expression did coincide with the sclerotome marker *Nkx3.1* in the caudal region (**new Fig. 5B**); then we used photoconversion of *Kaede*, and followed up the fate of the photoconverted cells (**new Fig. 5C-F**), which confirmed our previous data on the multiple fates of the somite ventral cluster cells.

Minor comments .

1. Some of the referencing and attribution of previous studies is sloppy.

For instance, two important papers from the Traver group are not referenced.

Kobayashi I, Kobayashi-Sun J, Kim AD, Pouget C, Fujita N, Suda T, Traver D. *Jam1a-Jam2a* interactions regulate haematopoietic stem cell fate through Notch signalling Nature. 2014 Aug 21;512(7514):319-23.

Clements WK, Kim AD, Ong KG, Moore JC, Lawson ND, Traver D. A somitic *Wnt16/Notch*

pathway specifies haematopoietic stem cells. *Nature*. 2011 Jun 8;474(7350):220-4. These are important papers that have primacy in the field for defining the way somitic signals regulate HSC formation and are not discussed in any detail.

- These papers related more to a direct, early molecular signaling from the somites to the angioblasts that will give rise to the HSPCs - a vast subject that also includes e.g. VEGF signaling (Leung et al. *Dev. Cell* 2013), whereas our study rather focuses on the participation of somite-derived migrating cells to the making of hematopoietic niches. When Clements et al. (2011) discovered that Wnt16/Dlc/Dld signaling within the somites was essential for HSPC emergence, they found that defects in this intra-somitic signaling also affected sclerotomal markers within the somite, which led them to suggest that sclerotome-derived cells such as vSMCs may deliver essential signals to the HSPCs-to-be (Clements & Traver, 2013). However their later study cited by the reviewer (Kobayashi 2014, confirmed by the last paper from the Kobayashi lab - Wada et al. *Development* 2022) led to a quite different interpretation, which no longer involved migrating sclerotome-derived mesenchymal cells, but rather the early Jam1a/Jam2a mediated adhesion and Notch mediated instructive interaction of Jam1a+ migrating angioblasts/future HSPCs with Jam2a+/Dlc+/Dld+ somite cells at the somite ventral surface (at a stage earlier than any migration of somite-derived cells). We have now summarized these differences in the Discussion, p.16.

Secondly the authors have mis-interpreted the results of another important study. Nguyen PD, Hollway GE, Sonntag C, Miles LB, Hall TE, Berger S, Fernandez KJ, Gurevich DB, Cole NJ, Alaei S, Ramialison M, Sutherland RL, Polo JM, Lieschke GJ, Currie PD Haematopoietic stem cell induction by somite-derived endothelial cells controlled by meox1. *Nature*. 2014 Aug 21;512(7514):314-8.

The authors contend two points about this paper: That this work shows that somite cells that colonise the dorsal aorta derive from the dermomyotome-derived cells and that their work contradicts these findings as their stromal cell, they contend, derive from the sclerotome. In actual fact Nguyen et al specifically show that these cells do not derive from cells that give rise the External cell layer the functional equivalent of the amniote dermomyotome. They instead derive from a somite compartment that expresses unique (non-sclerotome) markers that they show by lineage analyses to be distinct to External cell layer progenitors. In fact they form in a mutually exclusive manner. Nguyen et al term this compartment the "endotome" as they show by lineage analyses that they give rise to endothelial cells in the dorsal aorta. Clearly, based on the data in the two analyses, it is obvious that these two studies describe two different cells types (in fact the Nguyen et al study examines the DA and does not examine CHT dynamics in any detail).

- It is not so. Here we are primarily concerned with their use of the Pax3a:GFP transgenics, which we have used in our own study (*Tg(pax3a:EGFP)^{tl150}*, established by the Ingham lab). Nguyen et al. first stated that "pax3:GFP transgenic embryos which express GFP within the early somite, exhibit perduring GFP within the ECL and endothelial cells of the DA" (and indeed, the Currie lab has used Pax3a:GFP expression as a marker of the dermomyotome/ECL in several papers before and after this one). Yet later on in the paper, for their fate mapping and rescue experiments (starting from Fig. 3), they stated that the pax3a promoter is "endotome specific" and use Pax3a:GFP and other pax3a promoter based transgenics now to specifically mark the endotome, or ablate its derivatives, or rescue its functions.

From their live imaging of *Tg(pax3a:GFP)* embryos, they conclude that some Pax3a:GFP+ cells are endothelial cells within the DA, or PCV or ISVs, and others are "vascular-associated cells" ("VACs"). But our own higher-resolution imaging of this same region of the trunk shows that none of the Pax3a:GFP+ cells belong to the DA (or PCV); they are actually all "VACs". Their misinterpretation is due to the fact that many of these Pax3a:GFP+ VACs are in very close apposition to the DA, or PCV (see our **Fig. 7B-D, Supplementary Fig. 7A,B, and Movies 8 and 9**; Movie 8 is new Movie showing a z-scroll at 26 hpf, in which endothelial

cell contours are delineated by vascular-specific membrane-bound mCherry, further evidencing that all *pax3a:GFP+* cells are DA-associated, not part of the DA). Their lineage analyses suffer from the same confusion. They are fine in their principle except that the resulting cells are not endothelial, but all “VACs” (what we have called mesenchymal stromal cells), and what they call the endotome is actually the sclerotome. Indeed they interpret *Meox1* expression in Figs. 1y,z and S2m as labeling the ECL, and VACs (for the ventromedial staining), but this ventromedial staining is actually a typical sclerotome staining (*Meox1* actually is a known marker of both the ECL/dermomyotome and the sclerotome – see e.g. Tani et al 2020 or Lummertz da Rocha et al. 2022 cited by Reviewer 2). The same occurs for the other claimed marker of the endotome, a *cxcl12b* probe (stainings shown in Fig. 2a-j and Fig. S3a-f, which the authors themselves describe as “localized initially to a central region of newly formed somites” and “expressed within cells of the ventral aspect of the somites”). So what actually happens in the *choker* (*meox1*-deficient) mutant studied by Nguyen et al. is an expansion of the sclerotome compartment (or at least of its VAC derivatives) and a concomitant reduction of the ECL. Thus the data in that paper, once correctly re-interpreted, show that these sclerotome-derived DA-associated VACs stimulate the emergence of HSPCs from the DA (more VACs -> more HSPCs), in agreement with our own conclusion from our Figs. 7 and S7).

We have now expanded a bit on this clarification in our Discussion, **p.16** (even though not as much as here, as a critical analysis of the Nguyen paper is not the focus of our study).

2. There is clear overreach on the interpretation of the authors results. This is best typified by the statement at the end of the abstract:

“Thus the sclerotome contributes essential stromal cells for each of the key steps of developmental hematopoiesis, and likely is the embryological origin of most if not all mesenchymal stem/stromal cell found in non-cephalic tissues.”

Unfortunately there is simply no evidence in the submission to support this statement. No fate mapping at all is reported in this study let alone in the majority of non-cephalic tissues.

- This last part of the last sentence of the abstract was merely meant to exclude the cephalic tissues, in which any potential MSCs would rather most likely derive from the cranial neural crests. Anyway, we have now shortened this sentence to what we have demonstrated: “Thus the sclerotome contributes essential stromal cells for each of the key steps of developmental hematopoiesis”, even though the last sentence of an abstract is typically meant to expand the perspective. We have rather done that in the Discussion.

Also note that we have now added double in situ hybridization with the sclerotomal marker *nkx3.1* and *ola-twist1* based Kaede photoconversion experiments that further support the sclerotomal nature and diverse cellular fates of the somite ventral clusters that we have documented.

REVIEWERS' COMMENTS

Reviewer #1 (Remarks to the Author):

The authors have addressed all of the points we highlighted in our initial review. We believe this manuscript is well worthy of publication in Nature Communications.

Reviewer #3 (Remarks to the Author):

The authors considerably strengthen the manuscript in response to all the reviewer's concerns. In response to my particular concerns the addition of the fate mapping to complement the excellent high resolution timelapse imaging is a very welcome addition. Also my concerns over novelty have diminished with the strengthening of the loss of function approaches, with the following caveat, which the authors rightly argue are novel observations. I have the following remaining concerns:

1. Crisper Mutations. My original critique criticised the reliance on morpholinos. The authors have responded with supplementary data figure 5b which is a panel of relative superficial analysis on crispants for the genes under study. While crispants hint at true loss of function this can only be confirmed by mutations that pass the germline. The authors own data (supplementary data figure 5b) clearly show the crispant editing is only partial in injected embryos as would be expected. Likely this underlies the variability of the phenotype (supplementary data figure 5 e). The modest analyses in supplementary data figure 5 should be confirmed in germline passing, stably inherited, validated true loss of function mutations. As novelty rests on the validity of the gene specific function asserted in this study this is critical to do.

2. Referencing and discussion.

I note the inclusion and discussion of the work from the Traver group which broadens the context of the paper's findings. However, I remained puzzled over the interpretation of the results of Nguyen et al. as my recollection of the results of this paper are at odds with the interpretation the authors present. I have consequently gone back and reread this work.

The authors of this current submission state that "These cells are clearly the same as the somite-derived pax3a:eGFP+ that Nguyen et al.⁴⁸ showed to foster HSPC emergence, but that they interpreted as partly endothelial cells (of the DA and other vessels) and partly "vascular associated cells" ("VACs")."

In their rebuttal the authors suggest they are specifically referring to analyses using the Pax3 transgene to make this assertion.

Firstly, as stated in my original critique Nguyen et al. are primarily concerned with somite cells that colonise the DA during HSC induction and does not examine the CHT in any detail so there is no morphological rationale to compare these two cell populations in this current work.

Secondly, a careful reading of the Nguyen et al. reveals these authors do not use the Tg(pax3a:EGFP)il150 established by the Ingham lab to make their initial observation and ablations they actually use a different plasmid based construct of the pax3 gene they generated themselves as described in the methods "Tg(pax3a:GFP)pc7: The pax3a:GFP construct was generated using p5E pax3a38, pME GFP, p3E polyA and pDestTol2pA2." This has a different, very useful, pattern to the Ingham lab's transgenic line as described in this and previous publications from this group as it comes on early and stays on in migrating cells: In the paper they describe what they see using this line "Furthermore, pax3aGFP transgenic embryos, which express green fluorescent protein (GFP) within

the early somite, exhibit perduring GFP within the ECL and endothelial cells (ECs) of the DA (Fig. 2k-p, Extended Data Fig. 3m-o9, v-x9), with the majority of the pax3a-GFP-positive cells expressing cxcl12b (Fig. 2n-n9, arrows).” Note cxcl12b is not a marker of the sclerotome. So there is no direct point of comparison to make here in these two different set of reagents used to examine pax3a positive cells.

Thirdly the observations by Nguyen et al. using this unique Pax3a transgene are used only to set the stage for the three independent fate mapping strategies that confirm the endothelial contribution of cells of the somite to the DA. These fate mapping strategies are highly distinct to the small focused analyses this submitted work contributes which consists of cells emerging from the somite edge at the CHT (this is not a criticism its just different). Nguyen et al use two distinct whole of somite fate mapping strategy using the mesogenin transgene to undertake Kaede photo conversion and then CRElox mediate fate mapping. This is then followed by iontophoretic single cell fate mapping of the early somite (Figure 3 Nguyen et al.) To be honest this is perhaps the most extensive set of fate mapping experiments ever performed on any cell type in zebrafish. All show endothelial localisation in the DA of somite derived cells. For a simple expose of this fact the authors are refereed to Figure 3a-d’ which used mesogenin nuclear Kaede photoconversion of somite cells to reveal colocalisation of somite derived red photoconverted nuclei in cells marked by the endothelial marking transgene fli-GFP. Incidentally, Nguyen et also defined in these analyses (see panel 3d”) and others within the paper the existence somite derived vascular associated cells (VACs), a term first coined by these authors, stating that the “The dorsal aorta (DA) is colonized by endotome cells, which contribute to endothelial cells and a second set of cells termed vascular associated cells (VACs).” It would perhaps be collegiate to cite this paper when first referring to these cells in the submission, perhaps in the extended introduction suggested by other reviewer’s as the observation of VACs by Nguyen et al. has primacy in the field.

So the facts support that the zebrafish somite contributes two sets of cells: the endotome that contribute endothelial cells in the DA and regulate HSC emergence and the VACs which are the subject of this current submission.

The authors in this work nicely extend the description and function of VACs in the CHT and better define their origins in the sclerotome but they do not have any bearing on the origin and function of cells of the endotome. So as stated in the original critique the authors have clearly confused two distinct cell types in their discussion of their analyses. I think it needs to be addressed and corrected prior to manuscript proceeding further. However, I do entirely agree with the authors that this issue is not at all the focus of this study and it is not clear to me why it is raised at all. I suggest that the author remove the discussion of Nguyen et al from the discussion completely.

Response to the Reviewer #3 (in blue)

Reviewer #3 (Remarks to the Author):

1. Crisper Mutations. My original critique criticised the reliance on morpholinos. The authors have responded with supplementary data figure 5b which is a panel of relative superficial analyse on crispants for the genes understudy. While crispants hint at true loss of function this can only be confirmed by mutations that pass the germline. The authors own data (supplementary data figure 5b) clearly show the crispant editing is only partial in injected embryos as would be expected. Likely this underlies the variability of the phenotype (supplementary data figure 5 e). The modest analyses in supplementary data figure 5 should be confirmed in germline passing, stably inherited, validated true loss of function mutations. As novelty rests on the validity of the gene specific function asserted in this study this is critical to do.

– We understand your intention, but we would like to ask you to accept this content as such additional experiment would fall beyond the timing of re-revision. We have done everything possible in our crispant analysis using F0 generation and the results show the same phenotype as the MO injected embryos. We also provided data on the incidence of phenotypes to consider the possibility that phenotypic heterogeneity is due to the efficiency of the gene editing.

1.2. Referencing and discussion.

I note the inclusion and discussion of the work from the Traver group which broadens the context of the papers findings. However, I remained puzzled over the interpretation of the results of Nguyen et al. as my recollection of the results of this paper are at odds with the interpretation the authors present. I have consequently gone back and reread this work.

The authors of this current submission state that “These cells are clearly the same as the somite-derived pax3a:eGFP+ that Nguyen et al.48 showed to foster HSPC emergence, but that they interpreted as partly endothelial cells (of the DA and other vessels) and partly “vascular associated cells” (“VACs”).”

In their rebuttal the authors suggest they are specifically referring to analyses using the Pax3 transgene to make this assertion.

Firstly, as stated in my original critique Nguyen et al. are primarily concerned with somite cells that colonise the DA during HSC induction and does not examine the CHT in any detail so there is no morphological rational to compare these two cell populations in this current work.

- As we have clearly reported in our manuscript, and again emphasized in our previous response to this reviewer, we obviously did not compare the work of Nguyen et al. on the trunk/DA with our data on the CHT, but with our data on the trunk/DA presented in Figs. 7 and S6 and Movies 8 and 9. Furthermore, in the revised manuscript, we added to our previous imaging at 36 hpf additional images taken at 26 hpf so as to match as well as possible the stages examined by Nguyen et al.. For this we had to do some guess work, because the “26-somites” stage indicated in several of their figures (e.g. Fig. 3d-d” and f-f”) was clearly incorrect. The bright-field signal indeed shows a typical pattern of slanted lines in the DA, a well-known artefact due to circulating blood cells; as blood circulation only begins about an hour after the last (30th) somite is formed, i.e. by 25 hpf (at the standard temperature for zebrafish raising of 28.5° C), these images were necessarily taken past 25 hpf (whereas the 26-somites stage corresponds to 22 hpf). The same problem holds for their Fig. 2 n-p: here the DA and underlying PCV are too close to each other to be 26-somites; at that stage, the DA and PCV are farther from each other, due to the numerous primitive erythroblasts accumulated between the DA and PCV before circulation begins. Therefore we made our additional images (Movie 8) at 26 hpf.

Secondly, a careful reading of the Nguyen et al. reveals these authors do not use the Tg(pax3a:EGFP)^{il50} established by the Ingham lab to make their initial observation and ablations they actually use a different plasmid based construct of the pax3 gene they generated themselves as described in the methods “Tg(pax3a:GFP)^{pc7}: The pax3a:GFP construct was generated using p5E pax3a38, pME GFP, p3E polyA and pDestTol2pA2.” This has a different, very useful, pattern to the Ingham lab’s transgenic line as described in this and previous publications from this group as it comes on early and stays on in migrating cells: In the paper they describe what they see using this line “Furthermore, pax3aGFP transgenic embryos, which express green fluorescent protein (GFP) within the early somite, exhibit perduring GFP within the ECL and endothelial cells (ECs) of the DA (Fig. 2k–p, Extended Data Fig. 3m–o9, v–x9), with the majority of the pax3a-GFP-positive cells expressing cxcl12b (Fig. 2n–n9, arrows).”

- In their Methods section, Nguyen et al. mention that in this study they used both the Tg(pax3a:EGFP)^{il50} established by the Ingham lab, which they designate there as “TgBAC(pax3a:GFP)^{il50}”, and a newly created Tg(pax3a:GFP)^{pc7}. In the main text, there is no mention whatsoever of their use of two different pax3a:GFP lines. So when they mention pax3a:GFP embryos for the first time in the main text, in the sentence cited by the reviewer above, we can only conclude that they found no difference between the expression patterns of these two Tg lines. Since in their previous papers they always used the Ingham (il50) line, if their newly created line showed such a critically distinct pattern they would obviously introduce this new line as such in the text.

Even in the Figure legends, one cannot determine which of the two lines was used. One clue could have been that in the Methods section they designated Tg(pax3a:EGFP)^{il50} as “TgBAC(pax3a:GFP)^{il50}” and their own new line as “Tg(pax3a:GFP)^{pc7}”. But this doesn’t work either, since e.g. in Fig.3e-f’ they use both terms (without the allele name) to obviously designate the same embryos.

Note cxcl12b is not a marker of the sclerotome. So there is no direct point of comparison to make here in these two different set of reagents used to examine pax3a positive cells.

- Since cxcl12b is expressed in paraxial mesoderm at early stages of development, it is quite possible that its expression remains in trunk sclerotome for some time during subsequent development (we have actually observed cxcl12b expression by ISH at 24 hpf at the ventral side/sclerotome area of the caudal somites - our unpubl. data). The images provided by Nguyen et al. using their cxcl12b probe (Fig. 2a-d and Fig. S3a-f) do suggest a sclerotome-like expression pattern, which the authors themselves describe as “localized initially to a central region of newly formed somites” and “expressed within cells of the ventral aspect of the somites”). In any case, this point is not what we discuss in our present manuscript.

Thirdly the observations by Nguyen et al. using this unique Pax3a transgene are used only to set the stage for the three independent fate mapping strategies that confirm the endothelial contribution of cells of the somite to the DA. These fate mapping strategies are highly distinct to the small focussed analyses this submitted work contributes which consists of cells emerging from the somite edge at the CHT (this is not a criticism its just different). Nguyen et al use two distinct whole of somite fate mapping strategy using the mesogenin transgene to undertake Kaede photo conversion and then CRElox mediate fate mapping. This is then followed by iontophoretic single cell fate mapping of the early somite (Figure 3 Nguyen et al.) To be honest this is perhaps the most extensive set of fate mapping experiments ever performed on any cell type in zebrafish. All show endothelial localisation in the DA of somite derived cells. For a simple expose of this fact the authors are refereed to Figure 3a-d’ which used mesogenin nuclear Kaede photoconversion of somite cells to reveal colocalisation of

somite derived red photoconverted nuclei in cells marked by the endothelial marking transgene *fli-GFP*.

- We have nothing against a possible contribution of somite cells to the endothelium. What we say is that Nguyen et al. show no convincing evidence that such cells are *pax3a:GFP*⁺. And we clearly don't see this in our own higher-resolution imaging of these cells in this region.

In their Fig. 3a-d' cited by the Reviewer, there is no use of mesogenin but mere injection of *nlsKaede* mRNA in the whole embryo, followed by photoconversion in the anterior somite. While these images appear to support a somitic contribution to the endothelium, the photoconverted cells are not shown to express *pax3a:GFP*. In the next panels (Fig. 3e-f''), they do combine mesogenin-driven Kaede with *pax3a:GFP*, but beside the fact that the "26-somites" staging there is wrong (see our comment above), the resolution of these images is not high enough to discriminate whether the double-labelled cells are endothelial or endothelium-associated cells (please see our Movie 8). In fact, comparison with the bright-field signal in Fig. 3f shows that the four upper arrows point at cells that lie dorsal to the hypochord, hence lateral to the notochord, and not in the DA as claimed (we imaged *pax3a:GFP*⁺ cells at that same place – see our Fig. 7B-b, arrow). And the lower arrow points to a *GFP*⁺ cell the shape of which does not fit with an endothelial but with an endothelium-associated cell (see our own images, e.g. Fig. 7B-b, arrowhead).

The other images advocating endothelial nature of *pax3a:GFP*⁺ cells similarly suffer from insufficient resolution and/or overexposure. E.g. in Fig.2 o-p.; firstly the double-labelled cell pointed by the arrow is not associated with the DA as claimed, but with the dorsal wall of the underlying PCV; secondly, the overexposure of the red (endothelial) signal makes it impossible to discern that the green signal actually is in tight apposition to the red signal (see our own images – e.g. Fig. 7B and Movie 8) and not sunk into it, leading to the false interpretation of a double labelling.

Incidentally, Nguyen et also defined in these analyses (see panel 3d'') and others within the paper the existence somite derived vascular associated cells (VACs), a term first coined by these authors, stating that the "The dorsal aorta (DA) is colonized by endotome cells, which contribute to endothelial cells and a second set of cells termed vascular associated cells (VACs)." It would perhaps be collegiate to cite this paper when first referring to these cells in the submission, perhaps in the extended introduction suggested by other reviewer's as the observation of VACs by Nguyen et al. has primacy in the field.

- The first place in our manuscript where we refer to these DA/PCV associated cells is towards the end of the Results section, since we were led to image the trunk area as a consequence of our observation that the expression of *pdgfra* and *alcama* that we observed in ventral clusters (sclerotome) of the caudal somites actually extended to the the somites of the trunk. That is how we were led to find that in the trunk, somite-derived *pax3a:GFP*⁺ or *ET37:GFP*⁺ mesenchymal cells were also present, many of them intimately associated with the DA and PCV, and that when they were compromised by *Alcama* or *Pdgfra* deficiency, the emergence of HSPCs from the DA was also compromised. Following these observations, it was then natural and necessary that we then connect these results to those of Nguyen et al. in our Discussion.

So the facts support that the zebrafish somite contributes two sets of cells: the endotome that contribute endothelial cells in the DA and regulate HSC emergence and the VACs which are the subject of this current submission.

- While the study by Nguyen et al. shows somite-derived cells to stimulate HSPC emergence, it cannot discriminate between effects due to somite-derived putative endothelial cells and VACs, since both are augmented in *meox1* mutants. Based on our own analysis of this region, we guess that most if not all of the effect that they see is due to VACs.

The authors in this work nicely extend the description and function of VACs in the CHT and better define their origins in the sclerotome but they do not have any bearing on the origin and function of cells of the endotome. So as stated in the original critique the authors have clearly confused two distinct cell types in their discussion of their analyses.

- We obviously disagree on this point. While we have nothing in principle against a possible contribution of the somites to the endothelium, our point still is that we didn't find any *pax3a:GFP*⁺ cell contributing to the DA in the trunk, but only *pax3a:GFP*⁺ DA- and PCV-associated cells, often so intimately associated with the DA or PCV that it is understandably easy to confuse them with endothelial cells, and in fact unavoidable as soon as the fluorescence images are at insufficient resolution and overexposed, as in the Nguyen paper. Of note, we have been leading experts since 2007 in live-imaging the DA region, HSPC emergence from the DA and their subsequent entry into circulation via the underlying PCV; we have provided the high-resolution imaging of this process published to date (Kissa, Murayama et al. *Blood* 2008; Kissa & Herbomel, *Nature* 2010; Lancino et al. *eLife* 2017). So we know the DA region micro-anatomy and HSPC emergence process extremely well. In addition, our electron microscopy images (Kissa, Murayama et al. *Blood* 2008, Fig.4) showed the presence in the trunk of stromal mesenchymal cells analogous to those of the CHT (previously analyzed by EM in Murayama et al., *Immunity* 2006), in very tight apposition to both the DA ventral wall and underlying PCV dorsal wall, evidencing that distinguishing them from the endothelium via live confocal imaging would be quite challenging.

I think it needs to be addressed and corrected prior to manuscript proceeding further. However, I do entirely agree with the authors that this issue is not at all the focus of this study and it is not clear to me why it is raised at all. I suggest that the author remove the discussion of Nguyen et al from the discussion completely.

- The common conclusion of our study and theirs about somite-derived *pax3:GFP*⁺ cells enhancing HSPC emergence from the DA should definitely be addressed in our discussion; we have to mention their result, as it has primacy. On the other hand, we also need to mention our own findings about the histological nature of these cells. Our high-resolution imaging and expertise in the micro-anatomy of this area clarifies that they are not endothelial but all endothelium-associated, often very closely. Nevertheless we have amended our phrasing of this paragraph to express our disagreement in a more neutral way ("*In our own analysis, we found all *pax3a:GFP*⁺ cells to be VACs, part of them in very close apposition to the DA or PCV endothelium, ...*")